# Glacier response to Holocene warmth inferred from in situ [10]Be and [14]C bedrock analyses in Steingletscher's forefield (central Swiss Alps)

Irene Schimmelpfennig[1], Joerg M. Schaefer[2], Jennifer Lamp[2], Vincent Godard[1], Roseanne Schwartz[2], Edouard Bard[1], Thibault Tuna[1], Naki Akçar[3], Christian Schlüchter[3], Susan Zimmerman[4], ASTER Team[1]*

[1]Aix-Marseille Université, CNRS, Coll France, IRD, INRAE, CEREGE, Aix en Provence, France
[2]Lamont-Doherty Earth Observatory of Columbia University, Geochemistry, Palisades, NY 10964, USA
[3]Institute of Geological Sciences, University of Bern, Bern, Switzerland
[4]Center for Accelerator Mass Spectrometry, Lawrence Livermore National Laboratory, Livermore, CA 94550, USA
*A full list of the team members appears at the end of the paper.

*Correspondence to:* Irene Schimmelpfennig (schimmelpfennig@cerege.fr)

**Abstract.** Mid-latitude mountain glaciers sensitively respond to local summer temperature changes. Chronologies of past glacier fluctuations based on the investigation of glacial landforms therefore allows for a better understanding of natural climate variability at local scale, which is relevant for the assessment of the ongoing anthropogenic climate warming. In this study, we focus on the Holocene, the current interglacial of the last 11,700 years, which remains matter of dispute regarding its temperature evolution and underlying driving mechanisms. In particular, the nature and significance of the transition from the early to mid-Holocene and of the Holocene Thermal Maximum (HTM) are still debated. Here, we apply an emerging approach by combining in situ cosmogenic [10]Be moraine and [10]Be-[14]C bedrock dating from the same site, the forefield of Steingletscher (European Alps), and reconstruct the glacier's millennial recession and advance periods. The results suggest that subsequent to the final deglaciation at ~10 ka, the glacier was similar as or smaller than its 2000 CE extent for ~7 kyr. At ~3 ka, Steingletscher advanced to an extent slightly outside the maximum Little Ice Age (LIA) position, and experienced sizes until the 19th century that were mainly confined between the LIA and 2000 CE extents. These findings agree with existing Holocene glacier chronologies and proxy records of summer temperatures in the Alps, suggesting that glaciers throughout the region were similar as or even smaller than their 2000 CE extent for most of the Early and mid-Holocene. Although glaciers in the Alps are currently far from equilibrium with the accelerating anthropogenic warming, thus hindering a simple comparison of summer temperatures associated with modern and paleo glacier sizes, our findings imply that the summer temperatures during most of the Holocene, including the HTM, were similar to those at the end of the 20th century. Further investigations are necessary to refine the magnitude of warming and the potential HTM seasonality.

## 1 Introduction

Mountain glaciers in most glacierized regions of the world, such as the European Alps, are currently rapidly retreating in response to accelerating global warming, driven by human-induced greenhouse gas emissions into the atmosphere (IPCC 2007, 2013, in press). Small mountain glaciers are reliable indicators of regional climate variations on decadal to multi-millennial timescales, because their mass balance is sensitively controlled by variations of meteorological parameters, in particular summer temperature and precipitation (Oerlemans, 2005). Investigating past glacier behavior and the underlying regional climate variability provides the opportunity to better understand the natural driving mechanisms within Earth's climate system and to help quantify the anthropogenic contribution to the ongoing climate evolution (e.g. Roe et al., 2021).

The current interglacial Holocene followed the end of the last glacial period ~11,700 years ago and is characterized by moderate climate variations, including both colder-than-today and warmer phases (Mayewski et al., 2004; Wanner et al., 2008). In the northern mid- and high-latitudes, many studies provide evidence of several millennia of warm conditions during the early and mid-Holocene, generally referred to as the Holocene Thermal Maximum (HTM) (e.g. Renssen et al., 2009; Axford et al., 2013; Heiri et al., 2015; Kobashi et al., 2017). However, the occurrence of extended periods that were significantly

warmer than recent decades are still debated (e.g. Marcott et al. 2013; Marsicek et al., 2018; Affolter et al., 2019; Kaufman et al., 2020; Bova et al., 2021). The response of mountain glaciers to these Holocene warm periods remains unclear because records of when and how long mountain glaciers have receded to modern extents or beyond are still scarce and challenging, because much of the potential evidence is buried beneath ice.

The European Alps are one of the regions that are best documented in terms of Holocene glacier behavior (Ivy-Ochs et al., 2009; Solomina et al., 2015), but existing Holocene glacial chronologies are dominated by studies of moraines and thus large glacier extents that occurred during cold episodes (e.g. Schimmelpfennig et al., 2012, 2014; Moran et al., 2017; Le Roy et al., 2017; Protin et al., 2019, 2021; Braumann et al., 2020, 2021). Most of the existing constraints on the timing and amplitudes of glacier recessions come from discrete radiocarbon dates of sub-fossil wood and peat (e.g. Porter and Orombelli, 1985; Baroni

and Orombelli, 1996; Nicolussi and Patzelt, 2000; Hormes et al., 2001, 2006; Deline and Orombelli, 2005; Joerin et al., 2008; Nicolussi and Schlüchter, 2012; Le Roy et al., 2015), and only few studies provide records that characterize glacier extents during the majority of the Holocene, including periods of glacier retreat (Joerin et al., 2006; Luetscher et al., 2011; Badino et al., 2018).

A more recently developed and powerful approach to addressing the chronological reconstruction of millennial-scale Holocene

glacier retreat relies on the measurement of in situ cosmogenic $^{14}$C exposure dating in deglaciated bedrock, combined with in situ cosmogenic $^{10}$Be or other dating techniques, so far applied in only a few studies around the globe (Goehring et al., 2011 and Wirsig et al., 2016 in the Alps; Schweinsberg et al., 2018; Pendleton et al., 2019, Young et al., 2021 in the Greenland/Baffin region; Rand and Goehring, 2019 in Norway; Johnson et al., 2019 in Antarctica). This method provides the possibility to quantitatively derive the total duration that deglaciated bedrock has been exposed, i.e. was ice-free, and how long it was buried

beneath ice throughout the Holocene. The pioneering studies applying in situ $^{14}$C-$^{10}$Be exposure-burial dating showed that Rhône glacier, located in the central Swiss Alps, was smaller than its ~2005 CE extent for 6.4 ± 0.5 kyr, i.e. for the majority of the Holocene (Goehring et al., 2011, 2013). The challenge of this approach arises from the need of additional chronological constraints to determine the specific number and timing of recession periods, if they were interrupted by successive glacier advances.

In this study, we focus on Steingletscher, an often-visited, small mountain glacier in the central Swiss Alps (Fig. 1). Its evolution is well-monitored since the end of the 19$^{th}$ century showing that it has been constantly retreating, loosing >1000 m of its length since 1985 CE as a response to the modern climate warming (GLAMOS, 2020). Its forefield has been subject to various scientific studies, including investigations of the glacier's responses to Holocene cold episodes (King, 1974; Schimmelpfennig et al., 2014). Schimmelpfennig et al. (2014) mapped and $^{10}$Be-dated the Holocene moraines in the forefield

of Steingletscher (Fig. 1) providing evidence of several large glacier extents between the early Holocene and the end of the Little Ice Age (LIA, ~14$^{th}$ to 19$^{th}$ century). This existing knowledge on the glacier's advances and its high sensitivity to warming make it an ideal target to investigate the more difficult question: how did this glacier respond to extended warm periods during the Holocene? We therefore present here new measurements of in situ cosmogenic $^{14}$C and $^{10}$Be in bedrock recently deglaciated in front of Steingletscher, generally following the approach applied at nearby Rhône Glacier (Goehring

et al., 2011). We combine the new data on glacier recession with the previously published local Holocene moraine chronology, as well as with earlier published bracketing radiocarbon ages (King, 1974; Hormes et al., 2006) and historical and instrumental documentation of the recent glacier evolution. Our principal objectives are to (1) temporally constrain the Holocene intervals during which Steingletscher was at least as retracted as in modern times, and (2) evaluate whether the Steingletscher and Rhône Glacier retreat histories are individual records or rather represent regional glacier responses to warming climate phases. We

then put the result into the context of Holocene climate and glacier evolution in the Alps to test the significance of the HTM at regional scale.

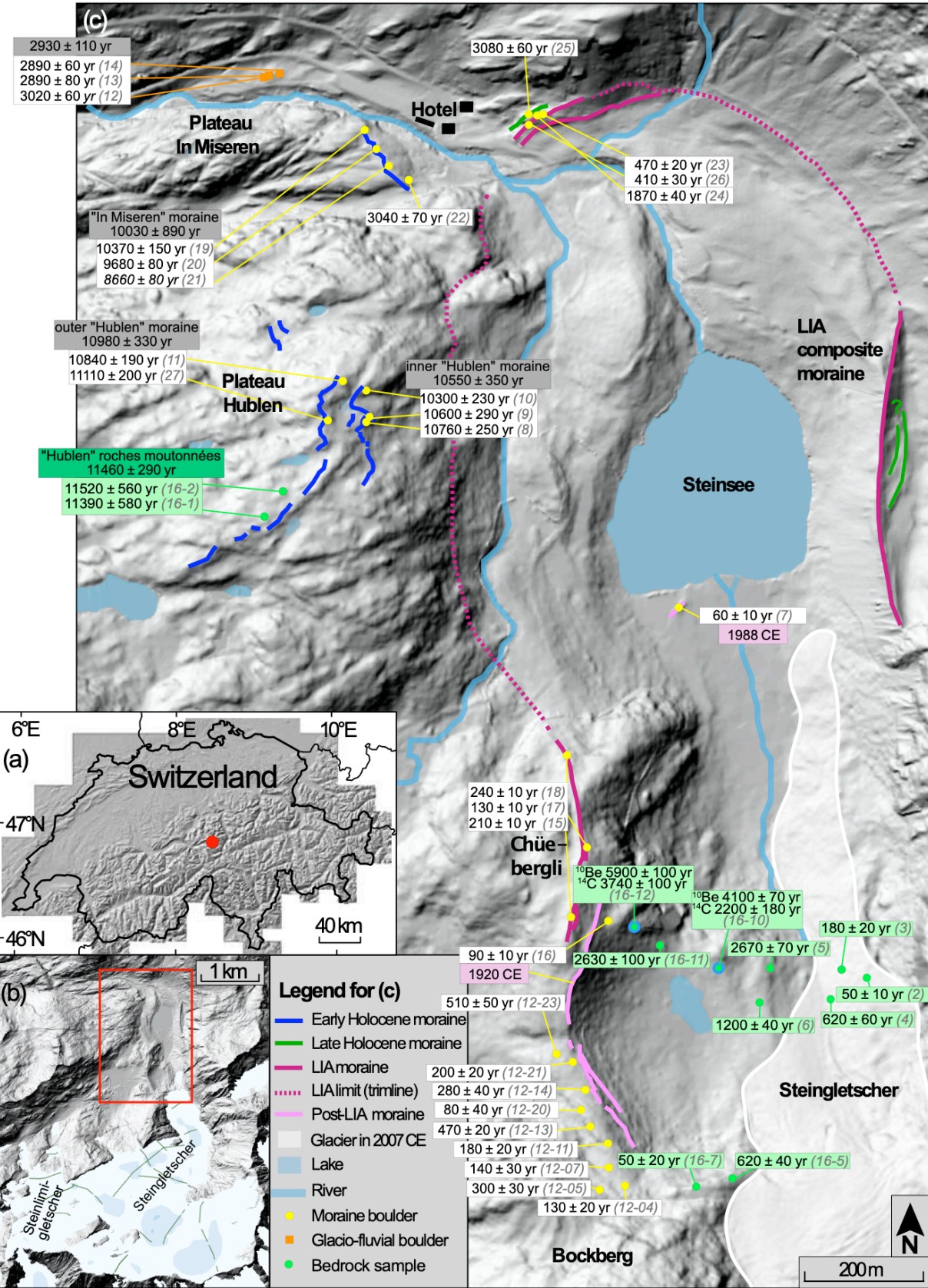

**Fig. 1: Maps of the study site based on shaded relief ALTI3D models by Swisstopo (https://map.geo.admin.ch). (a): Switzerland and the location of Steingletscher as a red dot (central Swiss Alps, 47°C). (b): Overview of the whole glacier catchment with the extents of Steingletscher and Steinlimigletscher in the year 2016. The red rectangular corresponds to panel (c). (c): Steingletscher's forefield with mapped Holocene moraines, their [10]Be exposure ages and 1σ analytical uncertainties (white boxes; recalculated from Schimmelpfennig et al., 2014; one outlier in italic) and the new bedrock sample locations with their apparent [10]Be and [14]C exposure durations and 1σ analytical uncertainties (green boxes). Mean landform ages are shown in darker boxes with 1σ uncertainties including analytical and [10]Be production rate uncertainties. Pink boxes give years of historically recorded moraine deposits.**

## 2 Study site, previous chronological work, and sampling strategy

Steingletscher is located in the eastern part of the Bernese Alps (~47°N, 8°E) at an altitude of 2220 m a.s.l. close to Susten Pass, and is part of a larger glacier catchment that hosts another glacier, Steinlimigletscher (Fig. 1b). It had a length of 3.35 km in 2019 CE and an area of 7.6 km$^2$ in 2013 (GLAMOS, 2019, 2020). Metagranitoids, gneisses and amphibolites constitute the geology of the glacier's surroundings. During the reference period 1991–2020 CE, the climate at the nearest weather station (Meiringen, 588 m a.s.l.) was characterized by an annual mean temperature of 8.7°C, a monthly mean temperature range between -1.1° C and 17.9° C, and an annual mean precipitation of 1341 mm (www.meteoswiss.admin.ch). During the reference period 1981–2010 CE, snow cover with a thickness of >50 cm occurred on 2.9 days (www.meteowiss.admin.ch; climsheet 2.1.6 / 05.01.2021). In the forefield of Steingletscher, the mean July temperature was 9.5°C between 1991 and 2020 CE (information provided by costumer service MeteoSwiss).

Steingletscher's forefield, stretching almost linearly towards the north, features glacially smoothed hills, moraines, trimlines and the proglacial lake Steinsee in the center of the half-bowl-shaped distal part of the forefield. The catchment's outlet is located in the north-western corner of this bowl and drains westward into Gadmen valley. On the left-lateral catchment flank, the up to 2090 m high "Plateau Hublen" and the lower "Plateau In Miseren" are characterized by glacially polished and lichen-covered bedrock knobs (*roches moutonnées*; Fig. 2a, b), interspersed with vegetated depressions, and small peat bogs and lakes (Fig. 1c). This landscape is overprinted by relicts of several moraine belts, the outer and the inner Hublen moraines, which were [10]Be dated at ~11.0 ka, ~10.6 ka and ~10.0 ka, from outer to inner (Schimmelpfennig et al., 2014). King (1974) obtained minimum radiocarbon ages for moraine formations from basal parts of peat bogs on Plateaus Hublen and In Miseren, which are in agreement with the [10]Be Holocene moraine data (Fig. 3a). They indicate that cold conditions still persisted during the deglaciation after the Younger Dryas (YD; 12.8–11.7 ka; Rasmussen et al., 2006). However, further evidence of YD related extents has not yet been identified. Here, we targeted two roches moutonnées located close to the highest point of Plateau Hublen, a few meters outboard of the outer Hublen moraine (Figs. 1, 2a, b) with the objective to date the timing of initial deglaciation of this plateau.

The absence of moraines after ~10 ka and throughout the mid-Holocene indicates warmer climate at that time (Schimmelpfennig et al., 2014). This is supported by radiocarbon ages of ~9 ka cal BP and from the mid-Holocene obtained from peat bogs on Plateau In Miseren (King, 1974; Table 2 in Schimmelpfennig et al., 2014; Fig. 3a). In addition, evidence of a substantially retracted mid-Holocene extent of Steingletscher comes from two wood fragments melted out from the glacier front between 1995 and 2000 CE and radiocarbon-dated at 5.3–4.8 ka cal BP and 4.8–4.6 ka cal BP (Hormes et al., 2006; Fig. 3a). In the same study, two organic silt samples from the forefield of the neighboring Steinlimigletscher, also collected between 1995 and 2000 CE, were radiocarbon-dated at 5.9–5.3 ka cal BP and 2.3–1.8 ka cal BP. Further information on the amplitude and duration of glacier recession during the early and mid-Holocene is missing, as geomorphic markers of glacier extents during that time were destroyed by the glacier re-advances during late Holocene cooling.

Evidence of late Holocene moraine formation comes from glacial boulders in the vicinity of the catchment outlet dated at ~2.9 ka (Schimmelpfennig et al., 2014). This maximum glacier extent during the late Holocene is corroborated by radiocarbon dates of organic material from soil and peat on the right-lateral side of the catchment outlet, which provide bracketing ages for the glacier advance and retreat around 3 ka ago (King, 1974; Table 2 in Schimmelpfennig et al., 2014; Fig. 3a). One large moraine boulder located on a ridge inboard of the ~3 ka moraine was [10]Be dated at ~1.9 ka and might represent a glacier extent at that time (Schimmelpfennig et al., 2014).

The most evident geomorphic markers of glacier expansion in Steingletscher's forefield are those from the Little Ice Age, including a sharp composite moraine on the eastern side of the forefield (Fig. 2d), multiple moraine ridges, and a clearly visible trimline (Figs. 2d, 4). Boulders from the preserved moraines yield [10]Be ages between ~570 and 140 years, consistent with the period of the LIA (Schimmelpfennig et al., 2014).

Historical and instrumental records provide constraints on glacial extents during the general retreat between 1850 CE and the beginning of the 21th century, which are here mainly based on previous compilations by King (1974) and Wirz (2007) and on glacier length measurements (GLAMOS, 2020). Fig. 3a shows the glacier outlines in the years 1850, 1920, 1933, 1973, 1988, 1999, and 2007. The length measurements of Steingletscher between the years 1893 and 2019 (Fig. 3b) indicate that the glacier had never retreated as much as in 2007 during that time. The most pronounced retreat of the 20th century occurred between 1960 and 1970, leading very briefly to a minimum extent that was comparable to that of the very beginning of the 21th century, but still slightly bigger than that in 2007 (Fig. 3b). During the subsequent advance of the 1970-1980's the glacier reached as far as into lake Steinsee. Since 1988, the glacier has been rapidly and continuously retreating and has lost >1000 m of its length in response to the modern global warming. Steingletscher is thus currently not in equilibrium with climate, but lags behind the accelerating warming by up to several decades.

To investigate how Steingletscher responded to naturally driven Holocene warmth, specifically how long it was similar in size as its modern configuration during the Holocene, we targeted two bedrock riegels for in situ $^{14}$C-$^{10}$Be exposure-burial dating. One riegel is located east of Chüebergli ("Chüebergli riegel" thereafter) and ~400 m long. The other riegel is located north of Bockberg ("Bockberg riegel" thereafter) and ~200 long. Both riegels are characterized by glacially polished, recently deglaciated roches moutonnées (Fig. 2c, d) that form steep cliffs towards the north. Chüebergli riegel was completely covered under at least ~140 m of ice during the LIA maximum, inferred from the altitude of the LIA composite moraine and trimline on the eastern catchment flank, and continued to be buried throughout most of the 20th century. Eight samples were collected on a transect on Chüebergli riegel from the highest and outmost bedrock surface down to the lowest bedrock surface in the glacial trough, following the sampling approach in Goehring et al. (2011). Note that the three lowest sample surfaces were still covered by ice in 2007 (Fig. 1c), but were ice-free by the sampling year 2010. From Bockberg riegel, two samples were analyzed (Fig. 1c).

After the glacier retreat in the 1960s, the outmost parts of the two riegels were temporarily ice-free, but ice-covered again in the 1970s–1980s by ~20 m of ice (Figs. 3a, 4). Chüebergli riegel might have been completely ice-free briefly around 1970. Both riegels were deglaciated in the early 21th century, and Steingletscher has continued to retreat since.

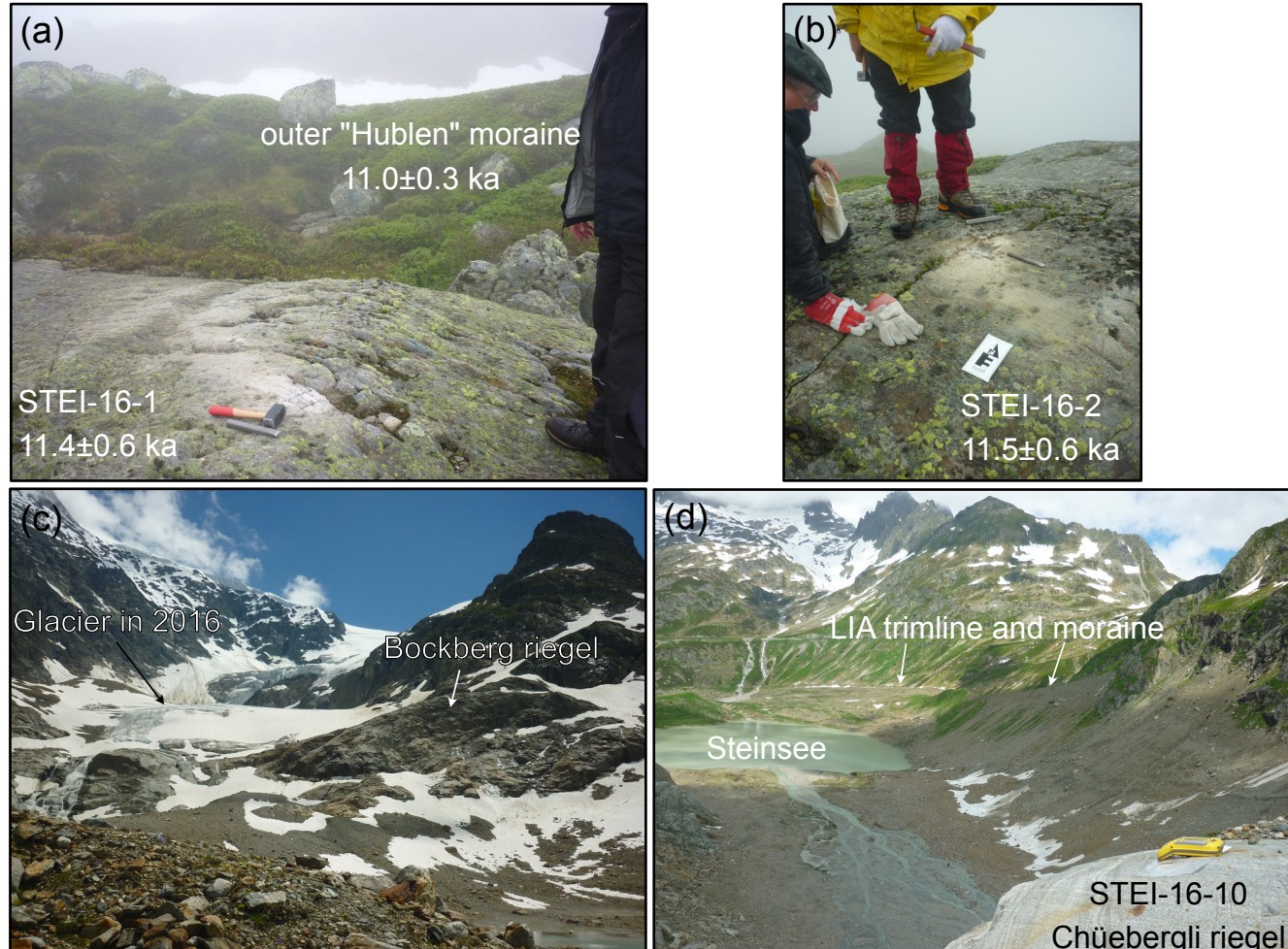

**Fig. 2: Photographs taken during field work. (a) and (b): Bedrock surfaces sampled close to the summit of Hublen Plateau. (c): Bockberg riegel and the glacier terminus in the year 2016. (d): Location of sample STEI-16-10 on Chüebergli riegel with the view on the eastern side of the Steingletscher forefield, highlightening the LIA trimline and composite moraine.**

160

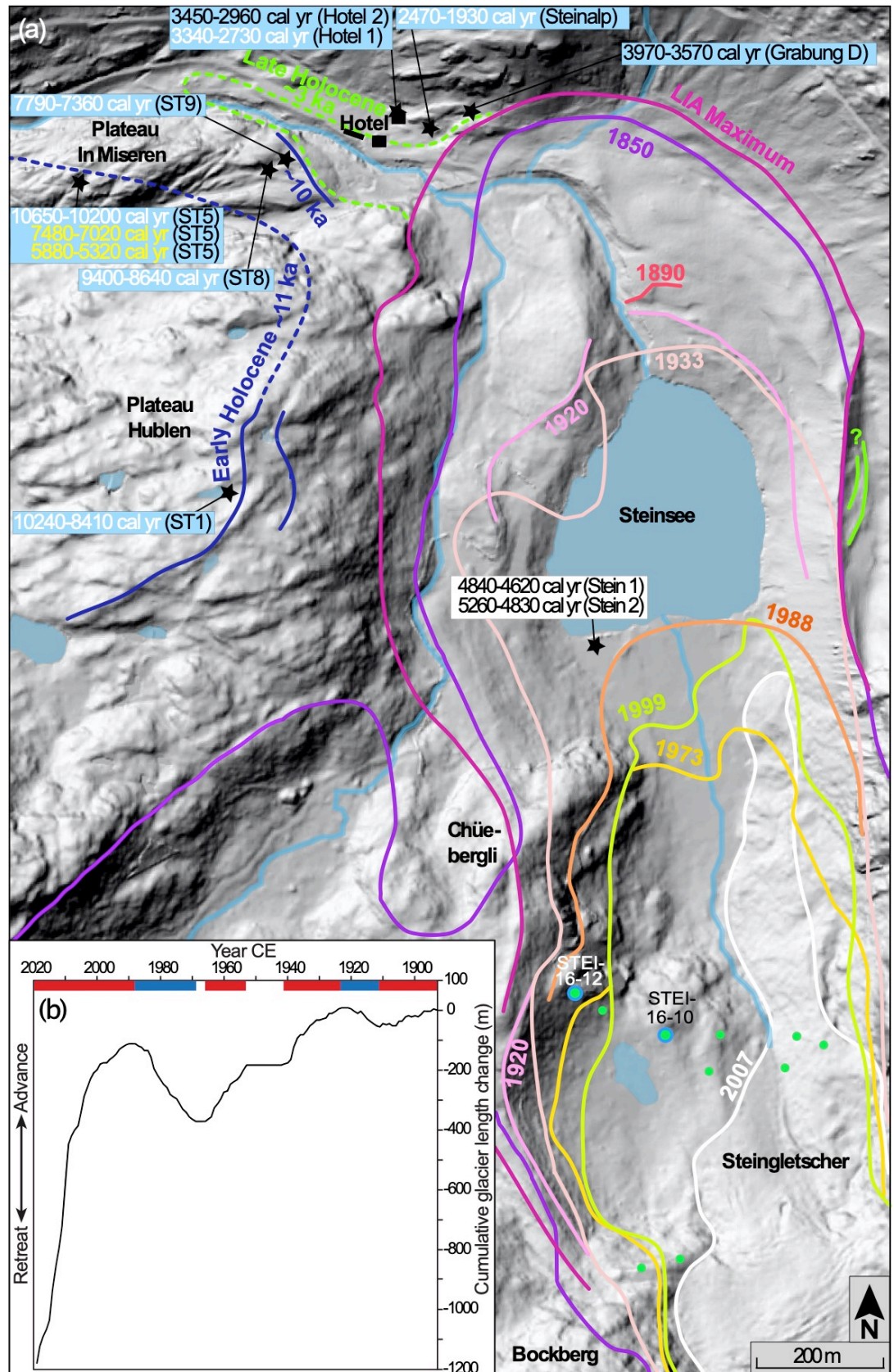

**Fig. 3: Map of Holocene extents of Steingletscher based on ¹⁰Be moraine dating (see Fig. 1c) for the period between ~11 ka and LIA, and on historical topographic maps and data from the Swiss Glacier Inventory (CH-INVGLAZ), adapted from Wirz (2007) and King (1974), for the period between 1850 and 1999 CE (a). Black stars represent locations of radiocarbon-dated organic material and corresponding calibrated ages from King (1974) (in blue boxes; black numbers are maximum ages and white numbers are minimum ages for moraine deposits or glacier retreat/advances; yellow ages correspond to certain pollen assemblages) and from Hormes et al. (2006) (in white box). Bedrock sample locations on Chüebergli and Bockberg riegels are shown as green dots, and samples STEI-16-10 and -12 are highlighted with blue rims. (b): Length measurements of Steingletscher since 1893 (from GLAMOS, 2020). Red and blue bars correspond to trends of glacier retreat and advance, respectively; intervals of glacier stagnation are in white.**

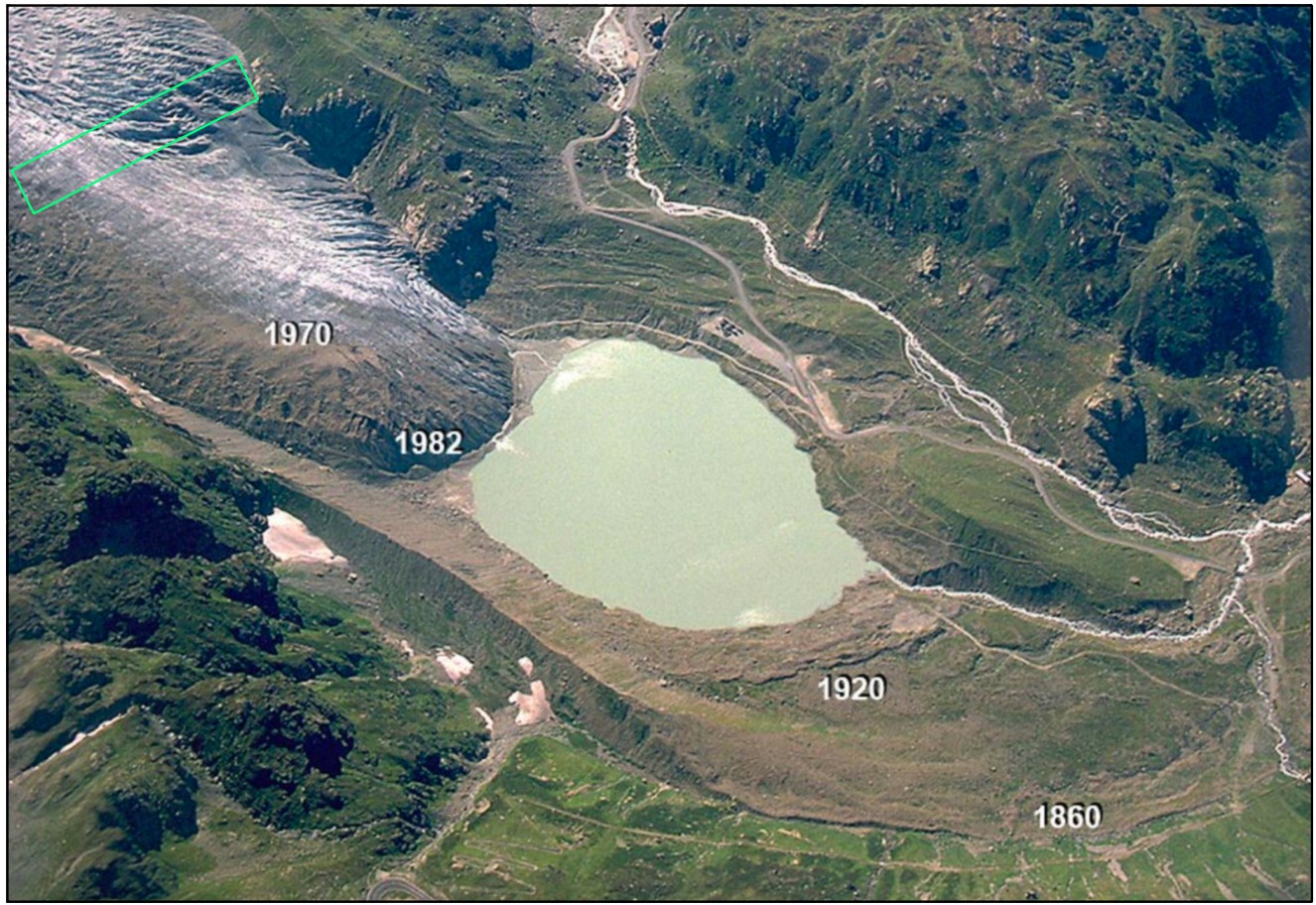

**Fig. 4: Aerial photograph of Steingletscher terminus position in 1982 CE with indications of the timing of earlier glacier extents, from www.swisseduc.ch. The location of the Chüebergli sample profile is indicated by the green box. Ice thickness in 1982 CE is estimated at ~50 m to ~20 m above the analyzed samples.**

## 3 Methodology

### 3.1 Fieldwork

The fourteen bedrock samples were collected during field campaigns in 2010 and 2016, targeting glacially polished and striated surfaces that were free of sediment cover. To minimize the risk of significant snow and sediment cover in the past, slightly sloping surfaces were preferred (Fig. 2d). Rock surface pieces with average thicknesses of ~2 cm to ~4 cm (Table 1) were sampled using either hammer and chisel alone or in combination with a cordless angle grinder and a diamond blade, preferring the quartz-rich parts of the gneissic lithologies. Latitude, longitude and elevation at the sample locations were recorded with a Trimble GeoTX GPS, the data reduction was conducted using the WGS coordinate system relative to EGM896 Geoid (Table 1). Correction factors for the shielding by the surrounding topography and the strike and dip of the sampled surfaces were determined from measurements using a handheld inclinometer. These correction factors range from 0.87 to 1.0 (Table 1).

**Table 1: Sample field information and analytical $^{10}$Be data. Carrier solution "CEREGE" used at LN2C has a $^9$Be concentration of 3025 µg g$^{-1}$; $^{10}$Be/$^9$Be ratios measured at ASTER (Arnold et al., 2010) were normalized to in-house standard STD-11 with the $^{10}$Be/$^9$Be ratio of 1.191 ($\pm$0.013) $\times$ 10$^{-11}$ (Braucher et al., 2015). LDEO carrier solutions "5.1" and "5.2" have $^9$Be concentrations of 1024 µg g$^{-1}$ and 1031.78 µg g$^{-1}$, respectively; $^{10}$Be/$^9$Be ratios measured at CAMS were normalized to standard 07KNSTD with the $^{10}$Be/$^9$Be ratio 2.85 $\times$ 10$^{-12}$ (Nishiizumi et al., 2007). A $^{10}$Be half-life of 1.387 ($\pm$0.01) $\times$ 10$^6$ years was used (Chmeleff et al., 2010; Korschinek et al., 2010). Samples were corrected for chemistry blanks by subtracting the number of atoms $^{10}$Be in the corresponding blank from that of the sample.**

| Sample name | Latitude (°N) | Longitude (°E) | Elevation (m asl) | Thickness (cm) | Shielding factor | Chemistry lab/ AMS facility | AMS # | Quartz weight (g) | Carrier # | Mass of $^9$Be in carrier (mg) | $^{10}$Be/$^9$Be (x10$^{-15}$) | $^{10}$Be concentration (x10$^3$ atoms g$^{-1}$) |
|---|---|---|---|---|---|---|---|---|---|---|---|---|
| **Bedrock on Hublen plateau** | | | | | | | | | | | | |
| STEI-16-1 | 46.725007 | 8.423675 | 2088 | 2.80 | 1.0000 | LN2C/ASTER | BXRX | 23.67 | CEREGE | 0.30649 | 289±14 | 242±13 |
| STEI-16-2 | 46.725253 | 8.423416 | 2078 | 2.68 | 1.0000 | LN2C/ASTER | BXRY | 18.63 | CEREGE | 0.30861 | 229±11 | 243±12 |
| **Bedrock riegel east of Chüebergli (from outer to inner)** | | | | | | | | | | | | |
| STEI-16-12 | 46.718277 | 8.431499 | 2089 | 2.84 | 0.9702 | LDEO/CAMS | BE41969 | 15.0545 | LDEO 5.2 | 0.18956 | 138.2±2.6 | 116.0±2.2 |
| STEI-16-11 | 46.717940 | 8.432130 | 2055 | 2.13 | 0.9559 | LN2C/ASTER | CBBW | 37.7273 | CEREGE | 0.46295 | 61.2±2.3 | 48.6±1.9 |
| STEI-16-10 | 46.717755 | 8.433492 | 2055 | 1.91 | 0.9706 | LDEO/CAMS | BE41968 | 15.1235 | LDEO 5.2 | 0.18987 | 92.7±1.7 | 77.5±1.4 |
| STEI-6 | 46.717310 | 8.434230 | 2052 | 1.84 | 0.9800 | LDEO/CAMS | BE34690 | 29.7057 | LDEO 5.2 | 0.18688 | 60.4±1.6 | 25.15±0.69 |
| STEI-5 | 46.717710 | 8.434520 | 2034 | 3.87 | 0.9800 | LDEO/CAMS | BE34689 | 30.2976 | LDEO 5.2 | 0.187190 | 125.4±3.7 | 51.4±1.5 |
| STEI-4 | 46.717320 | 8.435030 | 2033 | 2.80 | 0.8729 | LDEO/CAMS | BE33301 | 76.5250 | LDEO 5.1 | 0.15442 | 84.3±1.4 | 11.2±1.1 |
| STEI-3 | 46.717630 | 8.435200 | 2031 | 4.07 | 0.9352 | LDEO/CAMS | BE33300 | 100.8085 | LDEO 5.1 | 0.15462 | 35.5±1.0 | 3.5±0.35 |
| STEI-2 | 46.717550 | 8.435590 | 1997 | 3.68 | 0.9352 | LDEO/CAMS | BE33299 | 80.0638 | LDEO 5.1 | 0.15534 | 8.18±0.67 | 0.972±0.097 |
| **Bedrock riegel north of Bockberg (from outer to inner)** | | | | | | | | | | | | |
| STEI-16-7 | 46.714159 | 8.432754 | 2170 | 2.33 | 0.9333 | LN2C/ASTER | CBBV | 49.4389 | CEREGE | 0.45647 | 3.87±0.44 | 1.18±0.33 |
| STEI-16-5 | 46.714319 | 8.433553 | 2158 | 3.14 | 0.9159 | LDEO/CAMS | BE41966 | 15.0289 | LDEO 5.2 | 0.18976 | 15.39±0.79 | 12.70±0.68 |

| Blanks | Processed with | | | | | | | | | | | Total number of atoms $^{10}$Be (x10$^3$ atoms) |
|---|---|---|---|---|---|---|---|---|---|---|---|---|
| BLC-3 | STEI-16-1/ -2 | | | | | LN2C/ASTER | BXSE | | CEREGE | 0.30444 | 9.7±1.3 | 196±27 |
| BLC-4 | STEI-16-7/ -11 | | | | | LN2C/ASTER | CBBQ | | CEREGE | 0.46146 | 1.93±0.28 | 59.6±8.8 |
| BLK1-2016Oct06 | STEI-16-5/ -10/ -12 | | | | | LDEO/CAMS | BE41970 | | LDEO 5.2 | 0.18904 | 0.34±0.10 | 4.3±1.3 |
| Blank_2_2013Jan11 | STEI-5/ -6 | | | | | LDEO/CAMS | BE34691 | | LDEO 5.2 | 0.18636 | 0.171±0.080 | 2.11±0.99 |
| Blank_1_2011Dec23 | STEI-2/ -3/ -4 | | | | | LDEO/CAMS | BE33298 | | LDEO 5.1 | 0.15462 | 0.95±0.15 | 9.7±1.6 |
| Blank_2_2011Dec23 | STEI-2/ -3/ -4 | | | | | LDEO/CAMS | BE33307 | | LDEO 5.1 | 0.15555 | 0.75±0.15 | 7.8±1.6 |

## 3.2 Analytical methods

We separated and decontaminated quartz from 14 samples, and extracted $^{10}$Be from the clean quartz after spiking with pure $^{9}$Be carrier (Table 1). Chemical processing was carried out at the Lamont-Doherty Earth Observatory (LDEO) Cosmogenic Nuclide Laboratory (New York, USA) according to the standard procedures described in Schaefer et al. (2009) and at the Laboratoire National des Nucléides Cosmogéniques (LN2C) at Centre Européen de recherche et d'Enseignement des Géosciences de l'Environnement (CEREGE, Aix en Provence, France) following routine methods described e.g. in Protin et

al. (2019). $^{10}$Be/$^{9}$Be ratios were measured at the Lawrence Livermore National Laboratory - Center for Accelerator Mass Spectrometry (LLNL-CAMS) and at Accélérateur pour les Sciences de la Terre, Environnement, Risques (ASTER) at CEREGE. All data related to the $^{10}$Be analyses are presented in Table 1.

In situ $^{14}$C was extracted from the quartz of the two bedrock samples from Chüebergli riegel with the highest $^{10}$Be concentrations (STEI-16-10 and -12). These extractions were performed at LDEO following the procedure described in

Goehring et al. (2014) and Lamp et al. (2019), with two updates: The purified $CO_2$ sample and blank gas were diluted with only small amounts of $^{14}$C-free gas corresponding to ~20 μg of C (Table 2); the sample and dilution gas mixtures were not converted into graphite, but sealed into pyrex break seals for $^{14}$C/$^{12}$C ratio measurements at the AixMICADAS facility at CEREGE using the ion source dedicated for gaseous samples (Bard et al., 2015, Tuna et al., 2018), thus avoiding the graphitization step. $^{14}$C concentrations were calculated following the method of Hippe and Lifton (2014) (Table 2).

### 3.3 Principles of the exposure-burial dating approach

Three types of glacial surfaces with different exposure histories are investigated in this study.

i) Moraine boulders: they can in most cases be assumed to have a simple exposure history, i.e. they were free from cosmogenic nuclides at the moment of their stabilization after the glacier had retreated from its advance and have since been continuously exposed. In this case, the analysis of $^{10}$Be alone usually provides the exposure age of the

surface. Based on the consistency of the $^{10}$Be moraine boulder ages of Schimmelpfennig et al. (2014), they seem to fulfill this condition.

ii) Bedrock surfaces that remained continuously ice-free during the Holocene: they are located outboard of the maximum Holocene glacier extent. We assume that they were covered long and subglacially eroded deep enough during the ~100 ka lasting last glacial period that ended with the YD, that these surfaces were free from

cosmogenic nuclides at the moment of their last deglaciation. Subsequently, they experienced a simple exposure history, if cover by sediment, soil or vegetation is negligible. The two dated bedrock samples from Hublen (STEI-16-1 and -2), located outboard of the outer Hublen moraine, are assumed to fulfill these conditions.

iii) Bedrock surfaces that were alternately ice-free and ice-covered during the Holocene: they are located inboard of all Holocene moraines. Like the bedrock type ii, their cosmogenic nuclide inventory is assumed to have been set

to zero during the last glacial period. If during the Holocene phases of no-glacier-cover were followed by glacier-cover during $10^2-10^3$ years periods and with moderate subglacial erosion, cosmogenic nuclide concentrations cumulated from several exposure periods but were reduced through the glacial erosion during ice-cover. Consequently, the analysis of $^{10}$Be alone in this type of sample provides a minimum duration of the cumulative exposure period. The 10 samples from Chüebergli and Bockberg riegels correspond to this type of bedrock.

In the case of the bedrock type iii, the combined analysis of $^{10}$Be and $^{14}$C allows for solving for the unknown exposure duration and erosion depth, if the moment of initial deglaciation can be constrained (see Sect. 3.5). This is because $^{14}$C (half-life 5.7 kyr) decays much faster than $^{10}$Be (half-life 1.39 Myr, Chmeleff et al., 2010; Korschinek et al., 2010), therefore their concentrations evolve differently as a function of exposure and burial: during exposure both nuclides accumulate at a nearly constant rate with the $^{14}$C/$^{10}$Be concentration ratio close to ~3; during burial under ice, production stops and only the decay of

the $^{14}$C concentration is notable on these relatively short time-scales leading to lower concentration ratios (Fig. 5; Hippe, 2017).

In addition, subglacial erosion of the initially exposed surface removes the superficial bedrock layers, thereby advecting less $^{14}$C and $^{10}$Be concentrated rock from depth to the surface (Fig. 5). If this subglacial erosion is negligible or largely dominated by abrasion, the $^{14}$C/$^{10}$Be concentration ratio is less than 3 in the collected samples (Fig. 5), while higher ratios indicate surface quarrying by the glacier (Rand and Goehring, 2019). The latter is because production at depth of $^{14}$C is higher relative to that of $^{10}$Be due to a higher $^{14}$C contribution from muons, which penetrate deeper under the subsurface than neutrons (Hippe, 2017). Thus, in the case of moderate erosion rates dominated by abrasion, the apparent (i.e. non-burial- and erosion-corrected) $^{10}$Be and $^{14}$C ages provide both minimum exposure durations, with apparent $^{14}$C ages being younger than their $^{10}$Be counter parts, and the $^{14}$C/$^{10}$Be concentration ratio less than ~3.

We make the assumption that during the periods of burial, the ice cover was always thick enough at our sample locations to hinder significant $^{14}$C accumulation via muogenic production. Shielding by >70 m of ice is required to reduce $^{14}$C production to 1% of its surface production, and under a thin ice cover of ~13 m $^{14}$C is produced at 10% compared to an ice-free surface, while $^{10}$Be is produced at only 1% (Hippe, 2017). The photograph in Fig. 4 allowed us to estimate that during the glacier extent in 1982, ice cover was on the order of ~20 m to 50 m above the sample locations, suggesting that $^{14}$C production in the subglacial rock surfaces during episodes of ice cover should be small enough to not significantly affect the interpretation of our data.

This exposure-burial bedrock dating approach thus allows us to determine the cumulative duration that the glacier retreated beyond the sample locations during the Holocene. As reference for this minimum amplitude of retreat, we refer to the extent in modern times when the glacier uncovered the sample locations for the last time, i.e. ~1999 CE for sample STEI-16-12 and ~2007 CE for sample STEI-16-10 (Fig. 3a). As the time elapsed between these two years is insignificant compared to the centennial to millennial time scales investigated here, we simplify and refer to 2000 CE for both sample locations.

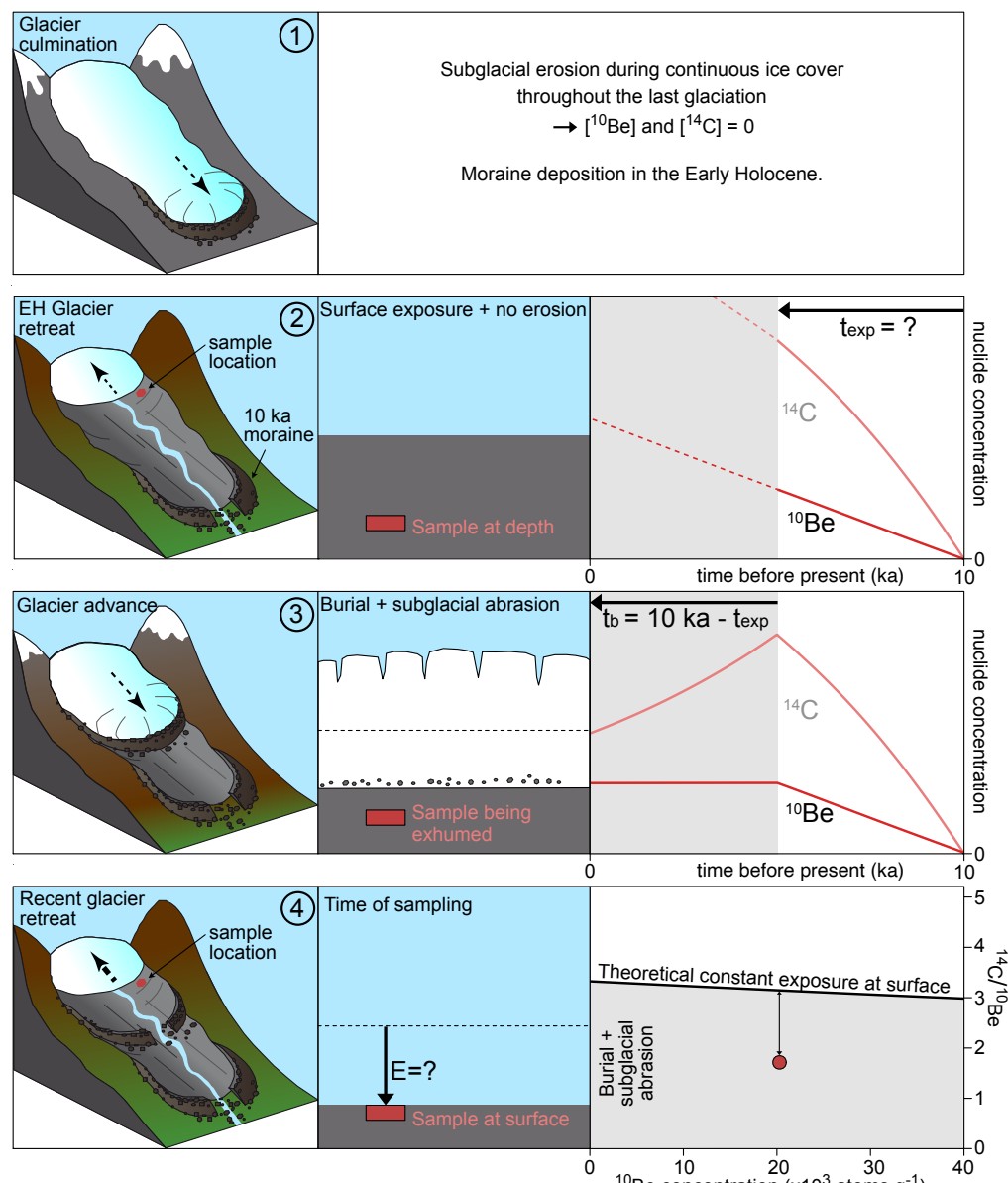

**Fig. 5: General concept of the in situ $^{14}$C-$^{10}$Be exposure-burial approach.** Left panels show a hypothetic scenario of alternating large and retracted glacier extents, covering and uncovering a fictive sample location (red spot). Middle panels depict the position of the sample material (red rectangle) with regard to the bedrock surface. Right panels of stages (2) and (3) illustrate the trajectories of the in situ $^{14}$C and $^{10}$Be concentrations in the sample during its exposure and advection (exhumation) towards the surface (Lagrangian perspective, see Sect. 3.5). Stage (1): Subsequent to continuous ice cover and deep subglacial erosion during the last glaciation, the bedrock is free from cosmogenic nuclides. Deposition of the Early Holocene (EH) moraines and subsequent glacier retreat constrains the beginning of the Holocene glacier retreat, i.e. 10 ka in this example. Stage (2): The sample location is exposed for an unknown duration $t_{exp}$, with the sample material remaining at the same depth (no surface erosion), where $^{14}$C and $^{10}$Be accumulate constantly. Stage (3): The sample location is buried by ice, which abrades the bedrock surface, thus progressively exhuming the sample material. No production of $^{14}$C and $^{10}$Be due to shielding by ice cover. Radioactive decay leads to steady loss of $^{14}$C (short half-life of 5.7 kyr), but does not affect the $^{10}$Be concentration due to the long half-life of this nuclide of 1.4 Myr. The burial duration $t_b$ is equal to the timing of initial EH glacier retreat minus $t_{exp}$. The dashed line represents the initially exposed surface elevation. Stage (4): Exposed since the recent glacier retreat, the sample material is now at the surface where it is collected. E represents the unknown erosion depth, i.e. the thickness of rock layer that was removed by abrasion during stage 3. The lowest right panel shows a two-isotope-plot with the thick black line representing continuous exposure ($^{14}$C/$^{10}$Be ratio = ~3) and the grey area burial and subglacial abrasion ($^{14}$C/$^{10}$Be ratio < 3). The $^{14}$C/$^{10}$Be ratio and $^{10}$Be concentration of the fictive sample is plotted as a red dot.

**Table 2: Analytical in situ $^{14}$C data. "Mass C (µg) sample" and "Mass C (µg) sample+diluted" are the masses of carbon released during extraction and after addition of $^{14}$C-free dilution gas, respectively. Oxalic acid OX-II is used as AMS standard, and all measurements are corrected for the AMS machine background blank with a $^{14}$C/$^{12}$C ratio of 0.0023. The percent of modern carbon value (pmC) includes the d$^{13}$C correction, which is undone following the calculations of Hippe and Lifton (2014). Samples were corrected for the procedural blank by subtracting the number of atoms $^{14}$C in the blank from that of the samples. The apparent $^{14}$C ages do not account for subglacial erosion effects and are therefore minimum ages, referenced to the sampling year 2016. See text for parameters used for the age calculations.**

| Sample name | AixMICADAS # | Year of measurement | Quartz weight (g) | Mass C (µg) sample | Mass C (µg) sample+diluted | pmC | Δ$^{13}$C (‰) | $^{14}$C concentration (x10$^3$ atoms g$^{-1}$) | $^{14}$C/$^{10}$Be conc. ratio | Apparent $^{14}$C exposure duration (kyr) "CREp" | Apparent $^{14}$C exposure duration (kyr) "former CRONUS" |
|---|---|---|---|---|---|---|---|---|---|---|---|
| **STEI-16-12** | 17003.2.1 | 2017 | 5.0101 | 98.5 ± 1.1 | 120.2 ± 1.4 | 15.35 ± 0.23 | -25.9 | 196.9 ± 4.1 | 1.70 ± 0.05 | 3.74 ± 0.30 (0.10) | 3.52 ± 0.24 (0.09) |
| **STEI-16-10** | 17003.3.1 | 2018 | 5.0475 | 27.0 ± 0.3 | 46.6 ± 0.54 | 26.22 ± 0.34 | -22.5 | 125.6 ± 2.5 | 1.62 ± 0.04 | 2.20 ± 0.18 (0.05) | 2.13 ± 0.13 (0.05) |
| **Blank** | | | | | | | | Total number of atoms $^{14}$C (x10$^3$ atoms) | | | |
| **BLANK_11-28-17_gas** | 17001.1.1 | 2017 | | 7.61 ± 0.09 | 28.6 ± 0.33 | 4.42 ± 0.16 | | 72.8 ± 2.8 | | | |

### 3.4 Calculations of simple exposure ages

The [10]Be moraine and apparent bedrock ages discussed below were calculated with CREp (http://crep.crpg.cnrs-nancy.fr/#/; Martin et al., 2017) choosing the local "Alpine" [10]Be spallation production rate (Claude et al., 2014) ($4.11 \pm 0.10$ atoms g$^{-1}$ yr$^{-1}$ as calculated in CREp via the link to the ICE-D calibration database), the ERA40 atmosphere model (Uppala et al., 2005), Lal-Stone time-corrected scaling (Lal, 1991; Stone, 2000; Balco et al., 2008) with atmospheric [10]Be-based VDM (Muscheler et al., 2005; Valet et al., 2005). The apparent [14]C bedrock ages were calculated accounting for spallogenic and muogenic production and radioactive decay, using the global [14]C spallation production rate of $12.22 \pm 0.89$ atoms g$^{-1}$ yr$^{-1}$, the muon parameters from Balco (2017) and the half-life of 5730 years. The same atmosphere model and scaling methods as for [10]Be were applied. A density of 2.7 g cm$^{-3}$ is assumed for all samples.

As most of the recent studies in the Alps used the former CRONUS-Earth calculator by Balco et al. (2008) to calculate cosmogenic nuclide ages, we also show the [10]Be moraine and apparent [10]Be and [14]C exposure durations of bedrock calculated with version 3 of this tool, choosing the same parameters as above regarding the [10]Be production rate (calculated as $4.04 \pm 0.38$ atoms g$^{-1}$ yr$^{-1}$ in the former CRONUS-Earth calculator using the calibration data from the ICE-D calibration database), atmosphere model and scaling scheme. Note that the calculator applies default parameters for the geomagnetic field model and the [14]C production rate.

All [10]Be and [14]C exposure ages and durations are shown in Tables 2 and 3. [10]Be exposure ages and durations calculated with both calculators differ by 1−2 % for early Holocene ages, and by at most 8 % for younger ages and exposure durations. [14]C exposure durations differ by 3−5 %. Higher differences in the results are due to high late Holocene inter- and intra-variability in the geomagnetic field records used in both calculators.

Unless otherwise stated, in situ cosmogenic ages corresponding to glacial surface types i) and ii) are reported in the text, tables and figures with the unit ka ("thousand years ago") with reference to 1950 CE (before BP), i.e. 60−66 years were deduced from their exposure ages. Exposure and burial durations of type iii) bedrock are given with the unit kyr and refer to their year of sampling (2010 or 2016 CE).

**Table 3: [10]Be moraine ages and apparent [10]Be bedrock ages at Steingletscher, with their full 1σ uncertainties. The uncertainties in parenthesis are the analytical 1σ uncertainties in the case of individual ages and standard deviations in the case of mean ages. Moraine ages are recalculated from Schimmelpfennig et al. (2014), using the same methods as for the bedrock samples (see Sect. 3.4). One outlier is in italics. The apparent [10]Be exposure ages do not account for subglacial erosion effects and are therefore minimum ages (with reference to the year of sampling). All moraine ages and bedrock ages from Hublen are referenced to 1950 CE.**

| Sample | [10]Be exposure age (ka BP) "CREp" | [10]Be exposure age (ka BP) "former CRONUS" |
|---|---|---|
| **Early Holocene - outer moraine on Hublen** | | |
| STEI-27 | 11.11 ± 0.32 (0.20) | 11.29 ± 1.05 (0.21) |
| STEI-11 | 10.84 ± 0.31 (0.19) | 11.00 ± 1.06 (0.21) |
| **Mean** | **10.98 ± 0.33 (0.19)** | **11.15 ± 1.07 (0.21)** |
| **Early Holocene - inner moraine on Hublen** | | |
| STEI-8 | 10.76 ± 0.36 (0.25) | 10.91 ± 1.07 (0.27) |
| STEI-9 | 10.60 ± 0.38 (0.29) | 10.75 ± 1.06 (0.31) |
| STEI-10 | 10.30 ± 0.33 (0.23) | 10.47 ± 1.02 (0.25) |
| **Mean** | **10.55 ± 0.35 (0.23)** | **10.71 ± 1.03 (0.22)** |
| **Early Holocene - moraine on "In Miseren"** | | |
| STEI-19 | 10.37 ± 0.28 (0.15) | 10.52 ± 1.01 (0.16) |
| STEI-20 | 9.68 ± 0.24 (0.08) | 9.84 ± 0.93 (0.09) |
| *STEI-21* | *8.66 ± 0.22 (0.08)* | *8.75 ± 0.83 (0.09)* |
| **Mean** | **10.03 ± 0.54 (0.49)** | **10.18 ± 1.07 (0.48)** |
| **Late Holocene moraine right of catchment outlet** | | |
| STEI-25 | 3.08 ± 0.10 (0.06) | 2.83 ± 0.28 (0.06) |
| **Late Holocene boulder - left of catchment outlet** | | |
| STEI-22 | 3.04 ± 0.10 (0.07) | 2.81 ± 0.28 (0.06) |
| **Late Holocene boulders - glacio-fluvial deposit** | | |
| STEI-12 | 3.02 ± 0.09 (0.06) | 2.79 ± 0.27 (0.05) |
| STEI-13 | 2.89 ± 0.10 (0.08) | 2.68 ± 0.27 (0.07) |
| STEI-14 | 2.89 ± 0.09 (0.06) | 2.69 ± 0.26 (0.05) |
| **Mean** | **2.93 ± 0.10 (0.08)** | **2.72 ± 0.26 (0.06)** |
| **LIA moraines right of catchment outlet** | | |
| STEI-23 | 0.47 ± 0.02 (0.02) | 0.46 ± 0.05 (0.01) |
| STEI-26 | 0.41 ± 0.03 (0.03) | 0.39 ± 0.05 (0.02) |
| STEI-24 | 1.87 ± 0.06 (0.04) | 1.76 ± 0.17 (0.04) |
| **LIA moraine on Chüebergli** | | |
| STEI-18 | 0.24 ± 0.02 (0.01) | 0.24 ± 0.03 (0.01) |
| STEI-15 | 0.21 ± 0.01 (0.01) | 0.21 ± 0.03 (0.01) |
| STEI-17 | 0.13 ± 0.01 (0.01) | 0.13 ± 0.02 (0.01) |
| **LIA moraines north of Bockberg** | | |
| STEI-12-23 | 0.51 ± 0.05 (0.05) | 0.50 ± 0.07 (0.05) |
| STEI-12-13 | 0.47 ± 0.03 (0.02) | 0.45 ± 0.05 (0.02) |
| STEI-12-05 | 0.30 ± 0.03 (0.03) | 0.30 ± 0.04 (0.02) |
| STEI-12-14 | 0.28 ± 0.04 (0.04) | 0.28 ± 0.05 (0.03) |
| STEI-12-21 | 0.20 ± 0.02 (0.02) | 0.20 ± 0.03 (0.02) |
| STEI-12-11 | 0.18 ± 0.02 (0.02) | 0.18 ± 0.03 (0.02) |
| STEI-12-07 | 0.14 ± 0.03 (0.03) | 0.14 ± 0.03 (0.03) |
| STEI-12-04 | 0.13 ± 0.03 (0.02) | 0.13 ± 0.03 (0.02) |
| STEI-12-20 | 0.08 ± 0.04 (0.04) | 0.08 ± 0.04 (0.03) |
| **Post-LIA moraine 1920** | | |
| STEI-16 | 0.09 ± 0.01 (0.01) | 0.10 ± 0.02 (0.01) |
| **Post-LIA moraine 1988** | | |
| STEI-7 | 0.06 ± 0.01 (0.01) | 0.06 ± 0.01 (0.01) |
| **Bedrock outmost position on Hublen** | | |
| STEI-16-1 | 11.39 ± 0.64 (0.58) | 11.56 ± 1.26 (0.60) |
| STEI-16-2 | 11.52 ± 0.62 (0.56) | 11.68 ± 1.25 (0.60) |
| **Mean** | **11.46 ± 0.29 (0.09)** | **11.62 ± 1.07 (0.08)** |
| | Apparent [10]Be exposure duration (kyr) "CREp" | Apparent [10]Be exposure duration (kyr) "former CRONUS" |
| **Bedrock riegel east of Chüebergli (from outer to inner)** | | |
| STEI-16-12 | 5.90 ± 0.17 (0.10) | 5.72 ± 0.55 (0.11) |
| STEI-16-11 | 2.63 ± 0.13 (0.10) | 2.45 ± 0.25 (0.10) |
| STEI-16-10 | 4.10 ± 0.12 (0.07) | 3.84 ± 0.37 (0.07) |
| STEI-6 | 1.20 ± 0.05 (0.04) | 1.19 ± 0.17 (0.03) |
| STEI-5 | 2.67 ± 0.10 (0.07) | 2.48 ± 0.24 (0.07) |
| STEI-4 | 0.62 ± 0.07 (0.06) | 0.61 ± 0.08 (0.06) |
| STEI-3 | 0.18 ± 0.02 (0.02) | 0.18 ± 0.03 (0.02) |
| STEI-2 | 0.05 ± 0.01 (0.01) | 0.05 ± 0.01 (0.01) |
| **Bedrock riegel north of Bockberg (from outer to inner)** | | |
| STEI-16-7 | 0.05 ± 0.02 (0.02) | 0.06 ± 0.02 (0.02) |
| STEI-16-5 | 0.62 ± 0.04 (0.03) | 0.61 ± 0.07 (0.03) |

## 3.5 Modelling of complex bedrock exposure history and erosion depth

Regarding the complex exposure history of bedrock type iii, three variables are unknown: the cumulative exposure duration $t_{exp}$, the cumulative burial duration $t_b$ and the erosion depth E, while the two-nuclide system allows for two of these unknowns to be solved (Goehring et al., 2011). In our study, the age of the youngest early Holocene moraine (In Miseren moraine, ~10 ka), provides the timing of initial deglaciation of Steingletscher's forefield, and therefore the cumulative burial duration can be constrained by $t_b = 10\ ka – t_{exp}$ (Fig. 5; Goehring et al., 2011). In the pioneer Rhône Glacier study by Goehring et al. (2011) the remaining two unknowns, $t_{exp}$ and E, were determined for each sample through the classical Eulerian computation, i.e. using the two equations for $^{10}$Be and $^{14}$C production and loss at the rock surface. In a follow-up study, Goehring et al. (2013) proposed in addition an isochron Bayesian approach that allows considering several samples simultaneously with the purpose to reduce the uncertainties for the whole data set.

Here, we use the Lagrangian instead of the Eulerian calculation method (Knudsen et al., 2019), combined with a Monte Carlo-Markov Chain inversion (MCMC) to constrain the values of $t_{exp}$ and E. We define a forward model to predict $^{10}$Be and $^{14}$C concentrations as a function of $t_{exp}$ and E in each of the two $^{10}$Be- and $^{14}$C-analyzed rock samples (STEI-16-10 and -12) during their advection toward the surface, as schematically illustrated in Fig. 5. For this, we apply the same production parameters and scaling method as for the simple exposure ages calculations. We assume that the $^{10}$Be and $^{14}$C concentrations were set to zero due to deep and sustained glacial erosion during the previous glacial phase (Fig. 5). The onset of the exposure history is set at 10 ka, as constrained by In Miseren moraine. The subsequent history is divided into two steps. First, the surface is exposed for a duration $t_{exp}$, leading to steady accumulation of $^{10}$Be and $^{14}$C. Second, the surface is covered by ice until present, leading to zero nuclide production, but bedrock erosion of a thickness E. The corresponding hypothetical $^{10}$Be and $^{14}$C concentration-time trajectories are shown in Fig. 5. We then use a standard MCMC approach to sample the parameter plane defined by $t_{exp}$ and E with the Metropolis–Hasting algorithm, and obtain the posterior distributions of these parameters for both samples. For each sample, we run 6 MCMC chains of length $10^5$, including a $10^3$ burn-in phase. For each parameter we report the average and standard deviation of the chain. The average erosion rate $\varepsilon$ is subsequently determined by $\varepsilon = E/t_b$.

The advantage of using the Lagrangian instead of the Eulerian approach is that it is more flexible when calculating the changing nuclide concentrations in a bedrock sample during its advection to the surface (Knudsen et al., 2019). In our specific case it conveniently allows considering the depth-dependent production and decay of $^{14}$C as a function of time, instead of summing up nuclide production and loss in a virtual surface sample. However, we stress that in our application, no significant differences result from the two approaches. Fig. 7l indeed depicts the concentration-time trajectories at the surface, i.e. using the Eulerian approach, both for the above-described single exposure-burial scenario and a more complex scenario with several exposure-burial alternations.

## 3.6 Recalibration of previously published radiocarbon dates

The radiocarbon ages previously published in King (1974) and Hormes et al. (2006) and discussed in this study were calibrated with the online program OxCal 4.4 (Bronk Ramsey, 2009), using its standard options and the IntCal20 calibration curve (Reimer et al., 2020) and are reported relative to the year 1950 CE. This changes the 2σ age intervals by at most 6% compared to the calibration with the earlier OxCal 3.9 version presented in Hormes et al. (2006). The radiocarbon ages published in King (1974) were uncalibrated and can therefore not be compared to the ages calibrated in our study. Note that with regard to the original studies, the general interpretations of all radiocarbon ages discussed here remain unaffected from the (re)calibration.

## 4 Results

Tables 2, 3 and Figs. 1c, 6a show the exposure ages and apparent exposure durations directly calculated from the measured $^{10}$Be and $^{14}$C concentrations. Also listed in Table 3 are all $^{10}$Be moraine boulder ages and moraine mean ages previously

published in Schimmelpfennig et al. (2014) and recalculated with the methods presented in Sect. 3.4. In the Tables, ages are shown with their full uncertainties (i.e. including analytical and production rate uncertainties) and analytical uncertainties only. In the text and in the figures, the individual ages are presented with their analytical uncertainties only, while moraine mean ages are presented in the text and in the figures with their full uncertainties, i.e. standard deviation and production rate error combined through simple error propagation (square root of the sum of their values in quadrature).

The two $^{10}$Be bedrock samples from Plateau Hublen yield indistinguishable ages of 11.39 ± 0.58 ka (STEI-16-1) and 11.52 ± 0.56 ka (STEI-16-2) with an arithmetic mean age of 11.5 ± 0.3 ka.

The two oldest apparent $^{10}$Be bedrock exposure durations from the Chüebergli riegel profile are 5.90 ± 0.10 kyr (STEI-16-12) and 4.10 ± 0.07 kyr (STEI-16-10). The other samples on this profile range between 2.67 ± 0.07 kyr (STEI-5) and 0.05 ± 0.01 kyr (STEI-2). All eight apparent $^{10}$Be ages from the profile present a general trend from high values at the outer margin towards

low values at the inner, lowest sample locations (Fig. 6a). This trend is consistent with the higher ice flow velocity in the center of a glacial trough that leads to deeper subglacial bedrock erosion during periods of ice-cover, thus reducing the $^{10}$Be surface concentrations at a higher rate (Goehring et al., 2011).

Only the two samples with the oldest apparent $^{10}$Be exposure durations (STEI-16-12 and STEI-16-10) were chosen for $^{14}$C analyses to ensure $^{14}$C measurements precise enough for a meaningful interpretation. The apparent $^{14}$C exposure durations of

the two analyzed samples are 3.74 ± 0.10 kyr and 2.20 ± 0.05 kyr, respectively, i.e. both are apparently younger than their $^{10}$Be counterparts. This trend and the low $^{14}$C/$^{10}$Be concentration ratios of 1.67 ± 0.05 and 1.62 ± 0.04, respectively, are consistent with temporary burial of the surfaces and indicate that subglacial erosion was moderate and dominated by abrasion (see Sect. 3.3; Table 2).

The two apparent $^{10}$Be bedrock exposure durations from the Bockberg riegel are 0.05 ± 0.02 kyr (STEI-16-7) and 0.62 ± 0.03

kyr (STEI-16-5). We decided to not use them for $^{14}$C analyses due to the low cosmogenic nuclide inventory. These samples are therefore only briefly discussed in the following text.

We highlight the exceptionally young apparent ages of samples STEI-2 and STEI-16-7, prepared/measured at LDEO/CAMS and LN2C/ASTER, respectively (Table 1), showing that at these facilities we are able to detect $^{10}$Be signals that correspond to ages as young as ~50 years with 20-40% analytical uncertainty.

Modelling of the $^{10}$Be and $^{14}$C data yields cumulative exposure durations $t_{exp}$ for STEI-16-12 and STEI-16-10 of 8.0 ± 0.3 kyr and 6.8 ± 0.3 kyr (1σ analytical uncertainties), burial durations $t_b$ of 2.1 ± 0.1 kyr and 3.2 ± 0.1 kyr, and subglacial erosion depths E (rates $\varepsilon$) of 21 ± 3 cm (*0.10 ± 0.02 mm yr$^{-1}$*) and 36 ± 3 cm (*0.11 ± 0.01 mm yr$^{-1}$*), respectively. Following the assumption that samples on a transect parallel to the ice flow experienced the same exposure history on millennial time scales (Goehring et al., 2011), an average ice-free duration and standard deviation of 7.4 ± 0.8 kyr can be deduced from the results

of STEI-16-12 and STEI-16-10. Given that STEI-16-12 lies ~160 m further outboard and 34 m higher in elevation than STEI-16-10, the longer exposure duration derived for STEI-16-12 could also either be due to shorter ice cover, or be an artifact of a thinner ice cover above this sample, the latter potentially leading to small muogenic $^{14}$C production in the subglacial surface of the sample (see Sect. 3.3). Distinct exposure histories of the two sample locations are explored in section 5.1.

Figure 6b shows the $t_{exp}$ and E distributions that can explain the observed concentration of one nuclide alone, $^{10}$Be or $^{14}$C,

resulting from the forward model (see Sect. 3.5) and represented by plain and dashed curves, respectively, for each sample (colored curves). The intersection of the two curves coincides with the average $t_{exp}$ and E values, determined with the MCMC inversion, thus providing a visual check of the optimal model solution. Following this strategy, we added the same type of curves obtained from the $^{10}$Be concentrations alone of four other samples from Chüebergli riegel (grey curves; not shown are the results from the two innermost samples due to their low $^{10}$Be concentrations). This approach allows evaluation of the range

of possible erosion depths and rates, illustrated by the yellow lines, even without combination with $^{14}$C analyses. Assuming that these four samples experienced a similar exposure-burial history as samples STEI-16-12 and STEI-16-10, their $^{10}$Be concentrations indicate subglacial erosion rates of >0.2 mm yr$^{-1}$ (Fig. 6b). This erosion rate estimate and the values inferred

from the combined [10]Be and [14]C data (~0.10–0.11 mm yr[-1]) agree with those from Rhône Glacier (~0.02–0.33 mm yr[-1]) and the steep Trift riegel (0–>2 mm yr[-1]) obtained with the same analytical method (Goehring et al., 2011; Steinemann et al., 2021).

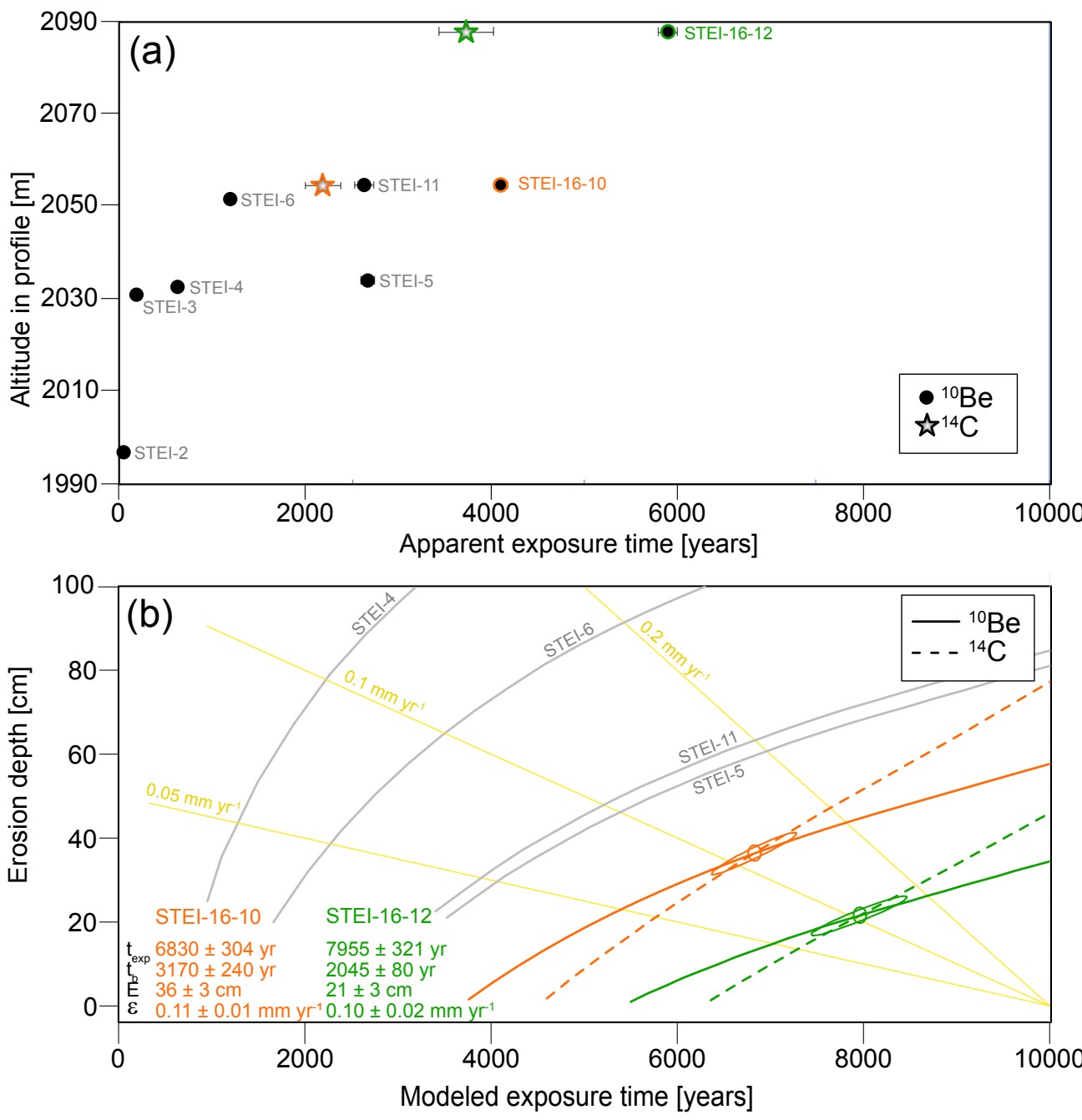

**Fig. 6: Exposure durations of bedrock samples inferred from measured in situ [10]Be and [14]C concentrations. (a): Apparent exposure durations calculated from [10]Be concentrations of 8 samples and [14]C concentrations of 2 samples as a function of position in glacial trough profile. The subglacial erosion effect during ice burial is not accounted for, calculations therefore yield minimum exposure durations. (b): Exposure durations and erosion depths modeled from combined in situ [10]Be and [14]C concentrations of samples STEI-16-10 and -12 (orange and green signatures). Solid grey curves represent possible exposure duration and erosion depth combinations for samples with [10]Be concentration only. Yellow lines indicate selected theoretical bedrock erosion rates.**

## 5 Discussion

### 5.1 Holocene timing and duration of Steingletscher retreat

Given the similarity in ages of the Hublen bedrock (highest and stratigraphically outmost position; 11.5 ± 0.3 ka) and the outer Hublen moraine (11.0 ± 0.3 ka), we infer that these bedrock surfaces experienced a simple exposure history and that their age

represents the timing of the first general deglaciation of the summit of Plateau Hublen at ~11.5 ka, during the transition from the YD to Holocene. This supports our assumption that the Hublen bedrock was free from significant [10]Be inheritance from earlier interglacials (see Sect. 3.3), thus providing evidence that this is the case for any bedrock surface further inboard in Steingletscher's forefield.

This date of deglaciation of the plateau summit also implies that the glacier had a greater extent prior to ~11.5 ka. However, chronological constraints on positions of Steingletscher prior to or during the YD do currently not exist.

Subsequent to the ~11.5 ka deglaciation of the Hublen Plateau summit, Steingletscher experienced a slow oscillatory retreat for about 1.5 kyr, similar to other glaciers in the Alps (Protin et al., 2021), and deposited the three preserved early Holocene moraines on Plateaus Hublen and In Miseren. Steingletscher's forefield was thus most likely completely covered by ice until the deposition of the In Miseren moraine (10.0 ± 0.9 ka). Chüebergli riegel was covered by at least 140 m of ice until then. Subsequently, the glacier considerably retreated, most probably for the first time in the Holocene. Although the amplitude of Steingletscher's recession at that time is unknown, we assume that Chüebergli and Bockberg riegels already became ice-free at ~10 ka. This is supported by radiocarbon-dated subfossil wood relicts that melted out of several glaciers across the Swiss Alps between 1990 and 2006 CE, indicating that the tree line had been well above the wood sample localities and that these glaciers were as small as or smaller between ~10 and ~8.2 cal BP than between 1990 and 2006 CE (Hormes et al., 2001, 2006; Joerin et al., 2006, 2008; Nicolussi and Schluechter, 2012).

Apart from two wood fragments collected in Steingletscher's forefield and radiocarbon-dated at 5.3–4.8 cal ka and 4.8–4.6 cal ka testifying to a glacier recession upstream of its ~2000 CE extent, no further direct constraints exist for the timing and amplitudes of Steingletscher's fluctuations during the mid-Holocene (Hormes et al., 2006; Fig. 7k). Evidence of recession of the neighboring Steinlimigletscher at 5.9–5.3 cal ka suggests that Steingletscher was in a retreated position, too, at that time (Hormes et al., 2006). During the late Holocene, the glacier considerably readvanced at ~3 ka ago and covered Chüebergli riegel again with ~140 m of ice or more. This happened again from at the latest 0.6 ka onward during the period of the LIA until the general retreat trend from 1850 CE on, which eventually uncovered Chüebergli riegel completely around 2000 CE.

Between ~3 ka and the LIA, the exact timing of further Steingletscher fluctuations is unknown. The bracketing radiocarbon age of a wood fragment of ~2.5–1.9 cal ka at the base of a ~1.10 m deep peat profile on fluvioglacial deposits near the LIA glacier limit points to glacier retreat (King, 1974). Significant glacier recession to modern extents at that time has also been documented at other Alpine glaciers, in particular in the detailed late Holocene records at Great Aletsch (Holzhauser et al., 2005) and Mer de Glace (Le Roy et al., 2015) (Fig. 7g, h). At Steinlimigletscher, one organic silt sample radiocarbon-dated at ~2.3–1.8 cal ka, attesting to glacier retreat beyond the ~2000 CE extent, supports a local impact of this probably regional-scale warming event (Hormes et al., 2006; Fig. 7k). A subsequent readvance of Steingletscher is tentatively suggested by one [10]Be boulder age of ~1.9 ka (Schimmelpfennig et al., 2014). Another period of considerable retreat can be assumed for the Medieval Warm Period (~1.3–0.7 ka), when several glaciers in the Alps are known to have retreated again to modern extents (Holzhauser et al., 2005; Le Roy et al., 2015; Fig. 7g, h).

Based on the [10]Be and [14]C data from Chüebergli riegel, we now explore the duration that the glacier had uncovered this area during the Holocene and put it into the context of the above described glacier fluctuations. The apparent [10]Be exposure durations from Chüebergli riegel alone already indicate that at least the outmost part of the profile has been exposed for a cumulative period of more than ~5.9 kyr during the Holocene, i.e. that the glacier was smaller for that duration than its extent in 2000 CE. The general trend of decreasing apparent [10]Be durations on the profile (by 2 orders of magnitude) indicates that the glacier temporarily covered and subglacially eroded the riegel again, otherwise similar apparent exposure durations would be expected for all samples on the profile. The *erosion-corrected* exposure durations modelled from the [10]Be-[14]C data of the two samples, indicate that the glacier was smaller than its 2000 CE extent for a total of ~7.4 kyr during the Holocene. Based on the assumptions and knowledge of relatively rare and short periods of glacier retreat after 3 ka ago, it is most likely that

almost all of this glacial retreat occurred between ~10 ka and ~3 ka. In Fig. 7l we propose two general scenarios of ice-free and ice-burial periods at Chüebergli riegel (i.e. glacier retreat and advance periods relative to 2000 CE) represented by theoretically reconstructed evolutions of $^{10}$Be and $^{14}$C concentrations at the surface of the locations where samples STEI-16-12 and -10 were collected. The simplest scenario, consistent with the measured concentrations of STEI-16-10, consists of a single ice-free period between 10 ka and 3 ka, and a continuously ice-covered period between ~3 ka and modern times (orange curves). In the more complex scenario, which yields final theoretical concentrations similar to the measured concentrations of STEI-16-12, the glacier retreat beyond the 2000 CE glacier extent occurred not only between 10 ka and 3 ka, but also during the late Holocene warm periods at 2.5−2 ka and 1.3−0.7 ka (green curves). The sum of the exposure durations in this scenario amounts to 8.1 kyr and is thus indistinguishable from the exposure duration of 8.0 ± 0.3 kyr analytically inferred from the $^{10}$Be and $^{14}$C measurements in sample STEI-16-12. For comparison, the scenario with a single exposure period between 10 ka and 2 ka is also shown for sample STEI-16-12 (also leading to consistent $^{10}$Be and $^{14}$C concentrations), which however is not consistent with the recorded glacier advance at ~3 ka and therefore appears less realistic.

Taken together, it is most likely that frequent glacier oscillations led to temporary burial of Chüebergli riegel, and that during the ~10 ka – 3 ka period, burial beneath ice occurred very rarely or not at all, while the ~3 ka to 2000 CE period was dominated by Steingletscher advances and ice cover of Chüebergli riegel.

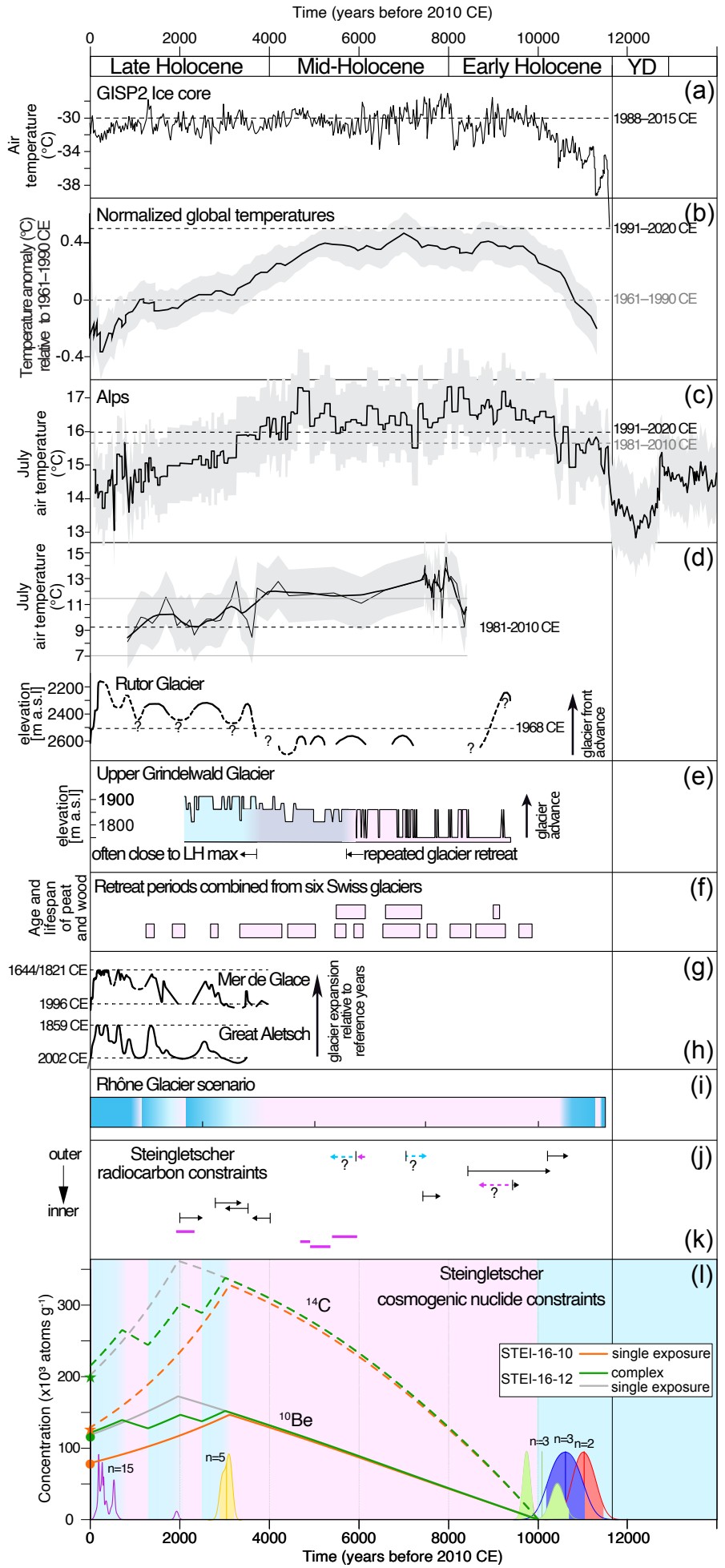

**Fig. 7: Holocene advance and retreat history of Steingletscher (lowest panel) in comparison with independent temperature and Alpine glacier records. (a): Greenland summit temperatures reconstructed from argon and nitrogen isotopes in GISP2 ice core (Kobashi et al., 2017). (b): Reconstruction of global temperature anomalies relative to the reference period 1961−1990 CE (grey dashed line) as presented in Marcott et al. (2013); uncertainty as grey band. The black dashed line represents the global annual mean temperature anomaly of the up-to-date reference period 1991−**
**2020 CE deduced from data provided by NASA (https://data.giss.nasa.gov/gistemp/). (c): Stacked summer temperature reconstructions at 1000 m a.s.l. based on Chironomid records from the Alpine region; uncertainty as grey band (Heiri et al., 2015). The dashed lines represent the averages of the modern July temperatures (1981–2010 CE and 1991–2020 CE) at the weather stations Meiringen (588 m a.s.l.) and Samedan (1708 m a.s.l.) (www.meteoswiss.admin.ch), extrapolated to 1000 m a.s.l., using a lapse rate of 6°C/1000 m as in Heiri et al. (2015). These two weather stations are**
**in close proximity to the two lakes Hinterburgsee and Stazersee, whose Chironomid records mainly contributed to the shown Holocene summer temperature reconstructions. (d): Summer temperature reconstruction based on pollen records from the forefield of Rutor Glacier (Italian Alps); uncertainty as grey band; modern July temperature (1981–2010 CE) at 2360 m in Rutor glacier forefield and uncertainty as horizontal lines. Fluctuations of the altitude of Rutor glacier front in comparison to 1968 CE (Badino et al., 2018). (e): Elevation changes of Upper Grindelwald Glacier**
**(central Swiss Alps) inferred from petrographic and stable isotope records in speleothems (Luetscher et al., 2011). (f): Recession periods of 6 Swiss glaciers based on radiocarbon-dated subfossil wood and peat (Joerin et al., 2006). (g) and (h): Reconstructions of glacier fluctuations of Great Aletsch (central Swiss Alps; length changes) and Mer de Glace (northern French Alps; altitude changes) based on radiocarbon-dating of subfossil wood and peat and archeological founds, historical data and dendrochronology (Holzhauser et al., 2005; Le Roy et al., 2015, respectively). The grey**
**extension added to the curve of Great Aletsch represents the loss in length until 2020 CE, based on measurement data (www.glamos.ch/en/factsheet#/B36-26). (i): Scenario of Holocene advance and retreat history of Rhône Glacier, central Swiss Alps, based on $^{10}$Be-$^{14}$C bedrock dating in combination with Alpine glacier advance records (Goehring et al., 2011). (j): Bracketing radiocarbon ages from peat bogs and outcrops at Steingletscher (King, 1974) with arrow lengths corresponding to the 2σ intervals of calibrated ages. Plain arrows represent bracketing ages for moraine deposits or**
**glacier advances. Dashed arrows correspond to climate cooling (blue) or warming (pink) trend interpreted from pollen assemblages. (k): Radiocarbon-dated wood fragments and organic silt samples melted out from Steingletscher and Steinlimigletscher between 1995 and 2000 CE, with line lengths corresponding to the 2σ intervals of calibrated ages (Hormes et al., 2006). (l): Steingletscher advance and retreat scenario inferred from cosmogenic nuclide constraints (this study). Mean $^{10}$Be moraine ages are represented by summed probability curves (from Schimmelpfennig et al.,**
**2014). Modeled evolutions of in situ $^{14}$C and $^{10}$Be concentrations for the 2 sample locations are shown by dashed and solid lines, respectively. Note that the trajectories of both nuclides at the bedrock surface are illustrated here (Eulerian perspective, see Sect. 3.5 for details). Blue bands are periods that are dominated by glacier positions larger than the glacial extent in ~2000 CE, pink bands are periods with dominantly smaller glacier extents.**

## 5.2 Comparison with Holocene glacier reconstructions in the European Alps

At a growing number of glacier sites, millennia-spanning records support the observed trend by providing evidence of significant and long-lasting glacier retreat during the early and mid-Holocene and progressive glacier re-advance during the late Holocene. Rutor Glacier in the western Italian Alps was smaller than its LIA maximum extent between 8.8 ka and 0.85 ka, and it was at least as contracted as in the 1960s CE between 8.8 ka and 3.7 ka (Badino et al., 2018; Porter and Orombelli, 1985; Fig. 7d). Upper Grindelwald Glacier experienced high-frequency elevation changes that were dominated by ice-loss

between 9.2 ka and 3.8 ka, followed by predominant advances close to the glacier's Holocene maximum (Luetscher et al., 2011; Fig. 7e). Radiocarbon-dated subfossil wood and peat fragments that had melted out of several retreating glaciers across Switzerland since 1990 CE, revealed frequent and prolonged periods of recession between ~10 ka and 3.5 ka followed by rare and short recessions during the late Holocene (Hormes et al., 2001; Joerin et al., 2006, 2008; Fig. 7f). Detailed reconstructions of late Holocene glacier fluctuations of Great Aletsch (central Swiss Alps) and Mer de Glace (northern French Alps), based

on radiocarbon-dated founds, historical data and dendrochronology, reveal frequent large advances from ~4-3 ka onward (Holzhauser et al., 2005; Le Roy et al., 2015) (Fig. 7g, h). Finally, Fig. 7i shows the Holocene retreat-advance scenario for Rhône Glacier (central Swiss Alps) inferred from combining the retreat duration of 6.4 ± 0.5 kyr with chronological data on glacier advances from the Alps (Goehring et al., 2011, 2013). The general consistency of the findings from Steingletscher with the Rhône Glacier scenario and with the findings from the other locations in the Alps validate the here applied approach and

confirms that the millennial retreat behavior of Steingletscher and Rhône Glacier represent regional glacial responses to Holocene climate warming.

Quantitative differences in the cumulative retreat durations inferred from the different methods seem to be consistent with the amplitude of glacier recession that is associated with the investigated sample material or location. While the in situ $^{14}C$-$^{10}Be$ dating approach suggests that glaciers were as retracted as the ~2000 CE glacier extents for ~7 kyr during the Holocene (Goehring et al., 2011; this study), the currently known recession periods inferred from the radiocarbon record by Joerin et al. (2006) add up to less, ~5 ka. The reason could be that the subfossil organic material implies glacier retreat significantly upstream of the sampling locations and the ~2000 CE glacier extents, as tree growth and peat development can only occur in deglaciated basins. This would roughly imply that since the final deglaciation (~10 ka), glaciers in the Alps were similar in size as in ~2000 CE during a total of ~2 kyr, while they were smaller than this extent for ~5 kyr. However, we acknowledge that this interpretation is tentative and will need to be verified, as the observed differences in the cumulative retreat durations might also be inherent to uncertainties in the dating approaches. In particular, it is likely that some periods of retracted glaciers are still unknown because the associated radiocarbon-dateable material has not yet been discovered. Another source for differing results from the two methods could also derive from unaccounted-for in situ $^{14}C$ production through thin ice (see Sect. 3.3). We also note that the existing data on Holocene glacier retreat does not allow verifying whether or not the glaciers completely vanished at some point during the Holocene.

### 5.3 Holocene glacier evolution in the Alps in the context of regional and global temperatures

Alpine summer temperature reconstructions based on chironomid and pollen assemblages are consistent with the Holocene glacier behavior in the Alps, showing a prolonged period of high temperatures during the early and mid-Holocene that might have been periodically and locally up to ~1-3°C warmer than in 1981-2010 CE (e.g. Heiri et al., 2015; Badino et al., 2018; Fig. 7c, d). Also, the proxy-based global mean temperature reconstructions by Marcott et al. (2013) and Kaufman et al. (2020) reveal a very similar trend suggesting an early to mid HTM followed by long-term cooling until the LIA (Fig. 7b). By contrast, model simulations of mean annual temperatures indicate that steady warming prevailed throughout the Holocene and that the recent decades are the warmest of the whole Holocene (Liu et al., 2014; Marsicek et al., 2018). Amongst the possible causes that have been proposed to explain these discrepancies, recent studies pay particular attention to the effect of seasonal biases (Marsicek et al., 2018; Affolter et al., 2019; Bova et al., 2021). According to this hypothesis, proxy-based global temperature reconstructions reflect warm-season, rather than annual temperatures, because the growth of biogenic proxies is controlled by summer temperatures.

As mid-latitude glacier records at regional scale are mainly driven by summer temperature evolutions (Oerlemans, 2005; Solomina et al., 2015), our results corroborate the existence of an extended warm-season HTM. The fact that alpine glaciers are currently out of equilibrium with the accelerating anthropogenic warming, lagging behind by up to several decades, complicates a direct comparison of summer temperatures associated with glacier positions of the Holocene and the Anthropocene. Glaciers in similar settings and of similar size as Steingletscher have response times on the order of a few to ~50 years (e.g. Oerlemans, 2012; Zekollari and Huybrechts, 2015), indicating that the summer temperatures responsible for Steingletscher's 2000 CE extent may have occurred in the middle or end of the 20th century, thus being 0.5-1°C less than in 2000 CE, according to the instrumental temperature record in the Alps (http://www.zamg.ac.at/histalp/; Auer et al., 2007). Our data therefore imply that summer climate during the HTM was similarly warm as or warmer than during the second part of the 20th century. No further inferences can be drawn on the amplitude of warming.

The occurrence of an extended warm-season HTM seems to support the hypothesis of seasonality during the early and mid-Holocene. However, the amplitude of this seasonality cannot be determined from glacier chronologies. Further investigations are therefore necessary to resolve the controversial annual temperature evolution of the Holocene. A recent reconstruction of seasonally unbiased temperatures in Greenland, based on argon and nitrogen isotopes, provides evidence of several early and mid-Holocene episodes (amounting to 27 % of the Holocene) with temperatures above the average of the 1988-2015 CE period

(Kobashi et al., 2017; Fig. 7a). This reconstruction is incompatible with the model-based steady annual warming and rather points to a hemispheric teleconnection with the trend of glacier fluctuations in the Alps.

In addition, we note that none of the continuous proxy records reveals a significant cooling event at ~3 ka that could explain the deposition of late Holocene moraines outboard of the LIA extent of Steingletscher. Hints of a LIA-like cooling at 4-3 ka are only noticeable in the pollen-based summer temperature record at Rutor Glacier (Fig. 7d). This inconsistency will also need to be further investigated, because several other moraines of similar age are preserved across the Alps (Schimmelpfennig et al., 2012; Le Roy et al., 2017; Moran et al., 2017), while other Alpine records indicate glacier extents at ~3 ka that are as

short as in ~2000 CE (Holzhauser et al., 2005; Le Roy et al., 2015; Fig. 7g, h).

Various drivers are relevant for Holocene climate change, i.e. external forcings at low (orbital summer insolation) and high frequency (volcanism and solar irradiance), feedback of the carbon cycle (greenhouse gases) and different climate boundary conditions linked to residual Northern Hemisphere ice-sheets (Mayewski et al., 2004; Wanner et al., 2008). Our findings of Alpine glacier retreats and advances are in line with the current understanding of Holocene climate change. Orbital summer

insolation modulates the long-term summer temperature evolution, thus driving millennial scale glacier evolution in the Northern mid and high latitudes (e.g. Solomina et al., 2015). Insolation is strongest in the Early Holocene followed by progressive decrease, consistent with glacier retreat during the Early and mid-Holocene and glacier re-expansion in the Late Holocene. Volcanic eruptions and changes in solar irradiance, superimpose centennial to decadal glacier fluctuations on the long-term trend during the Late Holocene (e.g. Büntgen et al., 2016; Jomelli et al., 2016). Finally, while greenhouse gas

concentrations were relatively stable over the Holocene, the accelerating anthropogenic greenhouse gas forcing has caused glaciers in the Alps and worldwide to retreat over the last century, with drastically increasing speed over the past few decades (Figs. 3b, 7g,h; e.g. Maurer et al., 2020; Roe et al., 2021; IPCC, 2007, 2013, in press). The high sensitivity of Steingletscher to the moderate summer temperature amplitudes during the Holocene implies that the glacier will continue to melt and shrink dramatically, and will most likely disappear if the human-induced warming is not reversed.

**6 Conclusions**

We find that Steingletscher responded highly sensitively to natural climate changes throughout the Holocene. It was as small as or smaller than its 2000 CE extent for a total of ~7.4 kyr throughout the Holocene. No later than ~10 ka, it shrank to its 2000 CE extent (or beyond) and advanced again to a LIA-like size at ~3 ka, followed by expanded extents throughout much

the past 3000 years until the rapid general retreat that started in the 19[th] century. This Steingletscher record is consistent with the regional Holocene glacier evolution in the Alps suggesting that glaciers across the Alps were as small as or smaller than their extents around ~2000 CE for most of the Holocene. The correlation between reconstructed summer temperature variability and the established glacier pattern demonstrates that the Alpine warm season temperatures between ~10 ka and ~3 ka, i.e. throughout the total of the HTM, were similar as or warmer than in recent decades. However, additional investigations

are needed to fully understand whether or not the early and mid-Holocene was characterized by significant seasonality.

Uncertainty also remains with regard to the amplitude of glacier recessions and thus the magnitude of warming. The exact amplitude of glacier retreat can indeed not be inferred from the here applied paired-nuclide approach, because the dated bedrock does not delineate a past glacier extent. As a perspective, applying this approach at various distances from the current glacier front could be a valuable strategy to add further spatial constraints on the amplitudes of glacier recessions, provided

subglacial bedrock erosion is low enough. In addition, the combination with complementary dating methods and glacier reconstruction approaches will help refine the long-term records both in terms of chronological and spatial constraints and thus add important knowledge to the Holocene glacier and climate picture in the region.

**Data availability**

All data will be availability in the ICE-D Alpine database (http://alpine.ice-d.org)

**Team List**

ASTER Team: Georges Aumaître, Didier Bourlès✝, Karim Keddadouche

**Author contribution**

IS and JS designed the study. IS, JS, NA, and CS conducted the fieldwork. IS, JL, RS, TT, SZ, and ASTER Team carried out the sample preparation and analyses. VG performed the modelling. IS wrote the manuscript with contributions from JS, JL,
VG, EB, NA and CS.

**Competing interests**

The authors declare that they have no conflict of interest.

**Acknowledgements**

IS is grateful for support by the French National Research Agency (project ANR-15-CE01-0007-01 WarHol). JMS gratefully
acknowledges support by the NSF grant # 1853881, the Columbia Climate Center, and the Vetlesen Foundation. We thank Jean Hanley (LDEO) and Magali Ermini (CEREGE) for assistance during sample preparation. The reviewers Yarrow Axford and Kurt Nicolussi provided constructive comments and suggestions that helped to improve the paper. We also thank Heinz Wanner and Martin Grosjean for relevant community comments, and Melaine Le Roy for suggestions that improved the geomorphic map. The ASTER AMS national facility (CEREGE) is supported by INSU/CNRS, IRD and ANR project ASTER-
CEREGE (program "projet thématiques d'excellence", action "Equipements d'excellence").

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
