# Peer review of "Glacier response to Holocene warmth inferred from in situ 10Be and 14C bedrock analyses in Steingletscher's forefield (central Swiss Alps)"

_Climate of the Past, 2021_

## Author Response (AR1)

POINT-BY-POINT REPLY TO REVIEWER AND COMMUNITY COMMENTS

All our replies are highlighted in blue.

RC1 : Yarrow Axford

This study builds upon previous published work at Steingletscher and Rhone glacier in the European Alps, adding paired 10Be- and 14C-derived apparent exposure durations from some key locations in the Steingletscher forefield.

The paper makes a significant new contribution to understanding the long-term history of glaciers in the European Alps, especially during past warm periods, and is well-suited to publication in COP. I find this to be an elegant study, shared in a well presented paper, and I don't have any major concerns or suggestions (which is unusual). The paper's introduction does a nice job setting up why the current approach is needed to fill in gaps in Holocene mountain glacier history. Methods and results are presented in good detail. I am not a cosmogenic isotope researcher, but am familiar with the study's methods and the study design (selection of sampling sites for 14C, inclusion of the outboard deglacial site to confirm erosional resettting of the cosmo clock, etc) seems strong. I am less able to rigorously evaluate the quantitative approach to modeling the two isotopes, but can say that the assumptions they apply seem solid and grounded in a diversity of available glacial geologic/geochronological evidence from past work. Section 3, notably 3.3, provides key summary background on the methodological approach. (I do suggest adding one conceptual figure to help broaden the reach of the paper; see my detailed comments below.) The figures in this manuscript are clear and informative, providing detailed information about the study sites and results.

**We thank Yarrow Axford for her positive and constructive comments. We agree with all her comments and suggestions and revised the manuscript according, which clearly strengthens the manuscript. Our answers are given below in bold.**

Minor comments (those most substantive are **ed):

The Abstract could use some tweaks to be more precise and impactful. Specifically:

Instead of "we apply a new approach" would it be more clear to describe this as an emerging or increasingly popular approach? Current wording made me think this study was the first such use of its 10Be-14C approach.

**We agree with this comment and have changed the wording accordingly: "we apply an emerging approach"**

Please clarify what is meant by "the predominant occurrence of glacier advances until the end of the Little Ice Age"

**We agree that this needs clarification. The text now reads "At ~3 ka, Steingletscher advanced to an extent slightly outside the maximum Little Ice Age (LIA) position, and experienced sizes until the 19th century that were mainly confined between the LIA and 2000 CE extents."**

The statement that "This implies that at least the summer climate of the HTM was warmer than that of the end of the 20th century for several millennia" requires that these glaciers have been roughly in equilibrium with climate of the late 20th Century rather than lagging far behind. I doubt this issue would have a big enough effect to nullify the quoted statement, but suggest discussing the assumptions of this conclusion more fully somewhere in the paper text to better support this somewhat provocative statement in the abstract.

**The reviewer is right that glaciers are currently not in equilibrium and that this needs to be discussed. We changed the text in the abstract as follows: "Although glaciers in the Alps are currently far from equilibrium with the accelerating anthropogenic warming, thus hindering a simple comparison of summer temperatures associated with glacier sizes, our findings imply that the summer temperatures during most of the Holocene, including the HTM, were similar to those at the end of the 20th century."**

**In Discussion section 5.3 we also added in lines 549-557: "The fact that alpine glaciers are currently out of equilibrium with the accelerating anthropogenic warming, lagging behind by up to several decades, complicates a direct comparison of summer temperatures associated with glacier positions of the Holocene and the Anthropocene. Glaciers in similar settings and of similar size as Steingletscher have response times on the order of a few to ~50 years (e.g. Oerlemans, 2012; Zekollari and Huybrechts, 2015), indicating that the summer temperatures responsible for Steingletscher's 2000 CE extent may have occurred in the middle or end of the 20th century, thus being 0.5-1°C less than in 2000 CE, according to the instrumental temperature record in the Alps (http://www.zamg.ac.at/histalp/; Auer et al., 2007). Our data therefore imply that summer climate during the HTM was similarly warm as or warmer than during the second part of the 20th century. No further inferences can be drawn on the amplitude of warming."**

Line 130 the word "century" is missing

**This is now corrected.**

Line 212 could be clarified, instead of "inboard of any of the Holocene glacier advances," how about "inboard of all Holocene moraines"? since evidence of some Holocene advances inboard of the moraines has been erased/covered.

**We agree and changed the text to "inboard of all Holocene moraines".**

**The conceptual model laid out in lines 220-233 is central to the paper, and well explained here – but things get complicated and hard to visualize when we get to section 3.5 and Figure 5b. An added conceptual figure illustrating the various relevant, hypothetical trajectories of the two isotopes would be very helpful in making this paper more meaningful for non-cosmo readers.

**We agree and have added a conceptual figure, now Fig. 5, that facilitates the understanding of the principles of the approach and the hypothetical trajectories of the two isotopes.**

Section 3.6. May be worth mentioning the typical size of change from the recalibration? I assume recalibration was undertaken to be thoroughly accurate but made only a small difference that doesn't affect conclusions.

**Yes, this is correct. Section 3.6 now reads: "The radiocarbon ages previously published in King (1974) and Hormes et al. (2006) and discussed in this study were calibrated with the online program OxCal 4.4 (Bronk Ramsey, 2009), using its standard options and the IntCal20 calibration curve (Reimer et al., 2020) and are reported relative to the year 1950 CE. This changes the 2σ age intervals by at most 6% compared to the calibration with the earlier OxCal 3.9 version presented in Hormes et al. (2006). The radiocarbon ages published in King (1974) were uncalibrated and can therefore not be compared to the ages calibrated in our study. Note that with regard to the original studies, the general interpretations of all radiocarbon ages discussed here remain unaffected from the (re)calibration."**

Lines ~465-469: Isn't it very likely that in the Joerin study some periods of retracted ice are simply not represented by discovered 14C-dateable deposits? That possibility is acknowledged in line 472 ("might also be…. lacking organic material from unknown retreat periods at the radiocarbon-dated sites") but it's not made to sound particularly likely, thus the need for the explanation about relative glacier size in lines 468-469. An "absence of evidence is not evidence of absence" scenario seems likely – but I don't know the Joerin study and may very well be missing something. Just clarify in the text.

**Yes, the reviewer is correct. We clarified the text in lines 529-535: "However, we acknowledge that this interpretation is tentative and will need to be verified, as the observed differences in the cumulative retreat durations might also be inherent to uncertainties in the dating approaches. In particular, it is likely that some periods of retracted glaciers are still unknown because the associated radiocarbon-dateable material has not yet been discovered. Another source for differing results from the two methods could also derive from unaccounted-for in situ $^{14}C$ production through thin ice (see Sect. 3.3). We also note that the existing data on Holocene glacier retreat does not allow verifying whether or not the glaciers completely vanished at some point during the Holocene."**

Line 474: do not capitalize chironomid

**This is now corrected.**

Line 491: clarify "steady ANNUAL warming". Interesting point about the Greenland N/Ar annual temp reconstruction contrasting with models showing annual warming through the Holocene.

**"Annual" has been added.**

Line 495: Cool point about widespread glacier advances ~3 ka but a lack of independent proxy evidence for temporary cooling to drive those advances. This seems like an interesting question/issue for glacial geologists and paleoclimatologists focused on the Holocene to ponder some more. Very very tenuously there may be hints of a corresponding climate event in the midge record in figure 6d?

**We agree and added in lines 566-567:" Hints of a LIA-like cooling at 4-3 ka are only noticeable in the pollen-based summer temperature record at Rutor Glacier (Fig. 7d). "**

**Line 500: "glaciers across the Alps were smaller than their modern extents for most of the Holocene" I think the take-home from this sentence would be even stronger if you put a timeframe on it. eg We find that Steingletscher was smaller than its present size for x-x kyrs in total throughout the Holocene, and given its expanded size throughout much of the past 3000 years, first shrank smaller than present no later than x ka. Likewise, line 503 could be more precise than "for several millennia of the HTM." (how many millennia and which ones?)

**Ok, we agree and have added the following text at the beginning of the conclusions: "We find that Steingletscher responded highly sensitively to natural climate changes throughout the Holocene. It was as small as or smaller than its 2000 CE extent for a total of ~7.4 kyr throughout the Holocene. No later than ~10 ka, it shrank to its 2000 CE extent (or beyond) and advanced again to a LIA-like size at ~3 ka, followed by expanded extents throughout much the past 3000 years until the rapid general retreat that started in the 19th century."**

**RC2: Kurt Nicolussi:**

This paper is an interesting and well written contribution to the topic of Holocene glacier evolution in the Alps. It is based on a so far rarely used approach to the analysis of Alpine glacier evolution and phases after the end of the Late Glacial. The study thus complements the usually temporally fragmentary evidence for Holocene glacier retreat phases in the Alps based on the analysis of wood and peat material. The approach as well as the analysis and modelling steps are presented in detail and - for me as a non-cosmogenic isotope researcher – largely comprehensibly, the discussion should undergo minor changes. Overall, the study is very well suited for publication in Climate of the Past.

**We are grateful to Kurt Nicolussi for his positive and helpful feedback that further improve the manuscript. We have thoroughly addressed all raised points, as detailed below in bold.**

Specific and minor points

Lines 336-338: It remains unclear why the two samples 16-10 and 16-12, analysed for both 10Be and 14C content, should have had the same exposure history and why the results were - therefore - averaged, as they come from clearly different positions in terms of sampling elevation. One site may have been covered by ice during glacier advances in the early to mid-Holocene, but the other may not have been reached, which could be one explanation for the different results. The inferred longer exposure time of sample 16-12, which was taken at a more elevated position, also points to such a scenario. A more complex consideration seems to make more sense here.

**We agree that it is likely that the two sample locations experienced a slightly different exposure history, because they are situated ~160 m in distance and ~34 m in elevation from each other. Unfortunately, based on our data, we cannot decide with certainty if the**

longer exposure duration inferred for sample 16-12 is due to a longer ice free duration or to the uncertainties inherent to the method (e.g. a thinner ice cover than above sample 16-10, see your later comment on the possible production of $^{14}$C through ice). In section 4 (results), we conservatively present the average exposure duration and standard deviation of 7.4 ± 0.8 kyr that can be deduced from the results of STEI-16-12 and STEI-16-10, following the simplified assumption that samples on a transect parallel to the ice flow experienced the same exposure history on millennial time scales (Goehring et al., 2011). Distinct exposure histories of the two sample locations are further explored in the discussion section 5.1. We changed the text according in lines 382-388: "Following the assumption that samples on a transect parallel to the ice flow experienced the same exposure history on millennial time scales (Goehring et al., 2011), an average ice-free duration and standard deviation of 7.4 ± 0.8 kyr can be deduced from the results of STEI-16-12 and STEI-16-10. Given that STEI-16-12 lies ~160 m further outboard and 34 m higher in elevation than STEI-16-10, the longer exposure duration derived for STEI-16-12 could also either be due to shorter ice cover, or be an artifact of a thinner ice cover above this sample, the latter potentially leading to small muogenic $^{14}$C production in the subglacial surface of the sample (see Sect. 3.3). Distinct exposure histories of the two sample locations are explored in section 5.1"

Line 385 It not clear to me why the find of a wood fragment points to a warmer climate. If it is about the inferred position of the tree line, which in turn is interpreted climatically, this would make sense.

We agree that this needs clarification. The wood fragment was found at the base of a peat profile on a fluvioglacial deposit near the LIA limits (King, 1974). It is rather the succession of the fluvioglacial deposit and the peat that indicate that the glacier retreated from a large extent at the timing given by the radiocarbon date of the wood (~2.5-1.9 cal ka). We changed the text in lines 435-437 to "The bracketing radiocarbon age of a wood fragment of ~2.5–1.9 cal ka at the base of a ~1.10 m deep peat profile on fluvioglacial deposits near the LIA glacier limit points to glacier retreat (King, 1974)."

Lines 394-417 These lines discuss the time span in which the Steingletscher was smaller than its 2000 CE extent, i.e. that the Chüebergli riegel was not ice-covered. ("... indicate that the glacier was smaller than its 2000 CE extent for a total of ~7.4 kyr during the Holocene"). Wouldn't it be also possible that - since a certain ice thickness (>70 m, lines 231-233) is required in each case to prevent 14C production - there were also less intense advances, that resulted in only minor ice overburden, but which clearly exceeded the 2000 CE extent? This would better explain the long exposure duration determined for sample 16-12 against the background of the state of knowledge on Holocene glacier evolution.

We agree that the text was not clear enough regarding the potential effect of ice cover on 14C production in a subglacial rock surface. We now clarified in lines 245-250: "Shielding by >70 m of ice is required to reduce $^{14}$C production to 1% of its surface production, and under a thin ice cover of ~13 m $^{14}$C is produced at 10% compared to an ice-free surface, while $^{10}$Be is produced at only 1% (Hippe, 2017). The photograph in Fig. 4 allowed us to estimate that during the glacier extent in 1982, ice cover was on the order of ~20 m to 50 m above the sample locations, suggesting that $^{14}$C production in the

**subglacial rock surfaces during episodes of ice cover should be small enough to not significantly affect the interpretation of our data.**

Line 476: The statement "HTM might have been ~1-3°C warmer than modern times" is quite strong, however, it needs a more specific temporal reference than "modern times" because of the currently fast changing climate conditions.

**We agree that the modern reference periods for the temperatures need to be specified, due to their rapid evolution. We changed the text to " Alpine summer temperature reconstructions based on chironomid and pollen assemblages are consistent with the Holocene glacier behavior in the Alps, showing a prolonged period of high temperatures during the early and mid-Holocene that might have been periodically and locally up to ~1-3°C warmer than in 1981-2010 CE (e.g. Heiri et al., 2015; Badino et al., 2018; Fig. 7c, d)."**

Lines 495-496 I agree that there is a lot of evidence for glacier advances and LIA-like extents between ca. 3.6 and 2.6 ka, but on the other hand, there is evidence for the glaciers Mer de Glace and Aletsch that they had a maximum extent as around 2000 CE at ca. 3 ka (see, e.g., Le Roy et al. 2015) - this should be added into this discussion.

**Ok, we agree. We changed the text in lines 567-570: "This inconsistency will also need to be further investigated, because several other moraines of similar age are preserved across the Alps (Schimmelpfennig et al., 2012; Le Roy et al., 2017; Moran et al., 2017), while other Alpine records indicate glacier extents at ~3 ka that are as short as in ~2000 CE (Holzhauser et al., 2005; Le Roy et al., 2015; Fig. 7g, h)."**

**Community comment by Martin Grosjean:**

I congratulate the authors for this exciting and comprehensive publication. I enjoyed reading it.

However, before final publication I strongly recommend revising the manuscript in the following points, mainly related to the Discussion and Conclusions:

**We are grateful for these constructive and relevant comments. We agree that taking into account the below three comments will further strengthen the paper and place the findings correctly in the context of the accelerating current glacier and climate change.**

- Please add in Fig. 6 b the global temperature for the reference period 1991 – 2020 which is significantly above 1961-1990; This is relevant to specify what 'today' means; 'today' should refer to 1991-2020 (recommendation WMO).

**We agree that the most up-to-date reference period should be used, given the rapid temperature rise. We added this information in the revised Fig. 7b. According to data available at the NASA website, the global mean surface temperatures in 1991-2020 was ~0.5°C above that in 1961-1990. We also added the mean July temperatures in the Alps for the reference periods 1981-2010 and 1991-2020 in Fig. 7c.**

- it would be helpful to show in Fig 6 g the extent of Mer de Glace and Great Aletsch in 2019/2020 (e.g. https://www.glamos.ch/en/factsheet#/B36-26 or WGMS data).

**Illustrating the strongly accelerating current glacier retreat is indeed relevant when studying glacier recession during the Holocene. The information has been added in the case of the graph depicting the length changes of Great Aletsch (now Fig. 7h). However, in the case of Mer de Glace, the graph taken from Le Roy et al. (2015) (now Fig. 7g) illustrates elevation changes and are thus not directly comparable to the available recent length measurements.**

- Seconding the Comment by Heinz Wanner: please add in the Discussion (Section 5.3) a short paragraph about the orbital forcing during the Holocene (particularly for summer) and how this influences summer TT and glacier lengths (based on the argument that glaciers are sensitive to summer temperature).

**We agree and added the following paragraph at the end of the discussion (lines 571-579): "Various drivers are relevant for Holocene climate change, i.e. external forcings at low (orbital summer insolation) and high frequency (volcanism and solar irradiance), feedback of the carbon cycle (greenhouse gases) and different climate boundary conditions linked to residual Northern Hemisphere ice-sheets (Mayewski et al., 2004; Wanner et al., 2008). Our findings of Alpine glacier retreats and advances are in line with the current understanding of Holocene climate change. Orbital summer insolation modulates the long-term summer temperature evolution, thus driving millennial scale glacier evolution in the Northern mid and high latitudes (e.g. Solomina et al., 2015). Insolation is strongest in the Early Holocene followed by progressive decrease, consistent with glacier retreat during the Early and mid-Holocene and glacier re-expansion in the Late Holocene. Volcanic eruptions and changes in solar irradiance, superimpose centennial to decadal glacier fluctuations on the long-term trend during the Late Holocene (e.g. Büntgen et al., 2016; Jomelli et al., 2016)."**

- I would also like to see a crystal clear statement (maybe in the Conclusions) that the main findings of this paper (small glaciers in the Early and Mid Holocene) is fully in line with the theory and current comprehensive understanding of Holocene climate change (including glacier variations) in the mid latitudes of the NH, and that recent glacier retreats (in the Alps and worldwide) and warming temperatures are undoubtedly attributable to anthropogenic forcing (e.g., Roe et al. 2021 The Cryosphere, 15, 1889–1905 and references therein; IPCC AR4, 5 and 6). The causes for (Early) Holocene glacier retreats were very different from those of today.

In light of recent glacier retreats under anthropogenic climate forcing (in the Alps, but also globally), it is most relevant to place Holocene glacier variations (this paper) and their causes in the appropriate, unambiguous and scientifically sound context.

**We agree. We added the following lines (579-584) at the end of the discussion: "Finally, while greenhouse gas concentrations were relatively stable over the Holocene, the accelerating anthropogenic greenhouse gas forcing has caused glaciers in the Alps and worldwide to retreat over the last century, with drastically increasing speed over the past few decades (Figs. 3b, 7g,h; e.g. Maurer et al., 2020; Roe et al., 2021; IPCC, 2007, 2013, in press). The high sensitivity of Steingletscher to the moderate summer temperature amplitudes during the Holocene implies that the glacier will continue to**

**melt and shrink dramatically, and will most likely disappear if the human-induced warming is not reversed."**

**Community comment by Heinz Wanner:**

The paper by Schimmelpfennig et al. provides an impressive overview of glacier fluctuations during the Holocene by analysing cosmogenic [10]Be in moraines and [10]Be-[14]C bedrock dating in the forefield of the Steingletscher in the central Swiss Alps. I recommend to add a small section, which explains the influence of the driving forcing factors. Basically, three different periods can be distinguished:

First, the early Holocene glacier retreats were strongly influenced by orbital forcing (Wanner et al. 2008, Solomina et al. 2015). Second, the advances and retreats of the last 4000 years were significantly influenced by groups of volcanic eruptions and solar irradiance changes (Bradley et al. 2016). During cooler periods with groups of volcanic eruptions as well as Grand Solar Minima, the glaciers often advanced. When the solar irradiance remained at medium levels and there were hardly any volcanic eruptions, it was likely warmer and the glaciers retreated. Typical cool periods with glacier advances are the Late Antique Little Ice Age (Büntgen et al. 2016) and the classical Little Ice Age (Grove 1988, Brönnimann et al. 2019). Thirdly, the massive glacier retreats of the present are clearly determined by anthropogenic forcing, which far exceeds the influence of solar irradiance.

**References**

Bradley R.S. et al., 2016. The medieval quiet period. The Holocene 26(6), 990-993.

Brönnimann S. et al., 2019. Last phase of the Little Ice Age forced by volcanic eruptions. Nature Geoscience. https://doi.org/10.1038/s41561-019-0402-y.

Büntgen U. et al., 2016. Cooling and societal change during the Late Antique Little Ice Age from 536 to around 660 AD. Nature Geoscience 9(3), 231-236.

Grove J.M., 1988. Little Ice Ages. Ancient and Modern. 2 Vols. Methuen, London, 718 S.

Solomina O.N., 2015. Holocene glacier fluctuations. Quaternary Science Reviews 111, 9-34.

Wanner H. et al., 2008. Mid- to Late Holocene climate change: An overview. Quaternary Science Reviews 27, 1791-1828.

**We greatly appreciate this constructive comment.**
**We agree and added the following paragraph at the end of the discussion (lines 571-579): "Various drivers are relevant for Holocene climate change, i.e. external forcings at low (orbital summer insolation) and high frequency (volcanism and solar irradiance), feedback of the carbon cycle (greenhouse gases) and different climate boundary conditions linked to residual Northern Hemisphere ice-sheets (Mayewski et al., 2004; Wanner et al., 2008). Our findings of Alpine glacier retreats and advances are in line with the current understanding of Holocene climate change. Orbital summer insolation modulates the long-term summer temperature evolution, thus driving millennial scale glacier evolution in the Northern mid and high latitudes (e.g. Solomina et al., 2015). Insolation is strongest in the Early Holocene followed by progressive decrease, consistent with glacier retreat during the Early and mid-Holocene and glacier re-expansion in the Late Holocene. Volcanic eruptions and changes in solar irradiance, superimpose centennial to decadal glacier fluctuations on the long-term trend during the Late Holocene (e.g. Büntgen et al., 2016; Jomelli et al., 2016)."**